# Understanding factors responsible for the slow decline of soil-transmitted helminthiasis following seven rounds of annual mass drug administration (2012–2018) among school children in endemic counties of Kenya: A mixed method study

**Janet Masaku**[1]*, **Collins Okoyo**[1], **Sylvie Araka**[1], **Rosemary Musuva**[2], **Elizabeth Njambi**[1], **Doris W. Njomo**[1], **Charles Mwandawiro**[1], **Sammy M. Njenga**[1]

**1** Eastern and Southern Africa Centre of International Parasite Control (ESACIPAC), Kenya Medical Research Institute (KEMRI), Nairobi, Kenya, **2** Centre for Global Health Research (CGHR) Kenya Medical Research Institute (KEMRI), Kisumu, Kenya

* mbinyamasaku@gmail.com, jmasaku@kemri.org

## Abstract

### Background

Soil-transmitted helminthiasis (STH) continue to be a significant health problem in Sub-Saharan Africa especially among school children. In Kenya, treatment of over five million children has been conducted annually in 28 endemic counties since the year 2012. However, the latest monitoring and evaluation (M&E) results indicated a slow decline of prevalence and intensity of STH in some counties after the seven rounds of annual mass drug administration (MDA). The current study sought to determine the factors associated with the slow decline in prevalence and intensity of STH among school children participating in the school deworming programme.

### Methodology

Mixed methods cross-sectional study was conducted in three endemic counties of Kenya. For quantitative technique, simple random sampling was used to select 1,874 school children from six purposively selected primary schools. The school children were interviewed, and a single stool collected and analysed using Kato-Katz technique. While for qualitative methods, 15 focus group discussions (FGDs) were conducted with purposively selected parents/guardians of school children. Data was collected through voice records using FGD and analyzed using NVIVO.

### Findings

Prevalence of any STH infection was 30.8% (95%CI: 28.7–32.9), with the highest prevalence observed in Vihiga County (40.7%; 95%CI: 37.4–44.4). Multivariable analysis

**Data Availability Statement:** All relevant data are within the manuscript.

**Funding:** The study received funding from the Kenya Medical Research Institute (KEMRI), Internal Research Grant (IRG): KEMRI/IRG/006/2017/2018 to JM. Website; www.Kemri.go.ke The funders had no role in study design, data collection and analysis, decision to publish, or preparation of the manuscript.

**Competing interests:** The authors have declared that no competing interests exist.

revealed that geographical location (OR = 3.78, (95%CI: 1.81–7.88) $p<0.001$), and not washing hands after defecation (OR = 1.91, (95%CI: 1.13–3.20) $p = 0.015$) were significantly associated with any STH infection. For qualitative analysis, majority of the parents/guardians of SAC felt that poor water sanitation and hygiene practices (WASH) both in school and household level could be a cause of continued STH infection. Also failing to include the rest of the community members in the MDAs were mentioned as possible contributors to observed slow decline of STH.

## Conclusions

There was moderate STH prevalence and mean intensity despite the seven rounds of repeated annual MDA. The study recommends a revamped awareness creation on WASH and community wide treatment.

## Author summary

Soil transmitted helminths (STH) are highly prevalent in most parts of Kenya especially among school age children (SAC). The National School-Based Deworming Programme (NSBDP) in Kenya was launched in 2012 by both Ministry of Health (MoH) and Ministry of Education (MoE) and has been offering mass drug administration (MDA) to 28 endemic counties in Kenya. The programme has been targeting all SAC, both enrolled and non-enrolled, as well as those in the pre-schools who live in sub-counties identified as having a high STH infection (prevalence above 20%). The deworming drugs are delivered by trained primary school teachers as recommended by world health organization (WHO). Approximately, over five million SAC and preschool age children (PSAC) have benefitted from the programme for over eight years. The country-wide infection reduced from 32.3% in 2012 to 12.9% in 2018, but varied prevalence rates observed in different counties. However, there are a few counties where the burden of STH has largely remained unchanged after the continued MDA. This study was conducted with a key focus to determine the factors which could be influencing the slow decline of STH infections among school children after continued MDA and give suggestions in future for similar programmes.

## Intorduction

Infections caused by soil-transmitted helminths (STH) are among the neglected tropical diseases (NTDs) targeted for elimination as public health problems in 2030 (EPHP) [1]. These infections continue to impose a great burden on populations with limited resources [2]. Globally the STHs; hookworms (*Ancylostoma duodenale* and *Necator americanus*), *Ascaris lumbricoides*, *Trichuris trichiura* and *Strongyloides stercoralis* infect approximately 1.5 billion people with majority of these infections occurring in Africa [3]. It is worth to note that *S. stercoralis* was not addressed in the current study. STH infection is a disease of poverty, commonly found among children due to poor personal and environmental hygiene, deprived living conditions, lack of water supply and impoverished sanitation. Chronic infections can have insidious effects on childhood development, including growth and cognitive development, while heavy infections may result in serious clinical disease [4]. The greatest burden of these infections is found

among school age children (SAC: 5–14 years) in endemic areas due to their high susceptibility to frequent exposure to contaminated environment especially when playing, eating unwashed fruits or raw vegetables, and bathing or drinking untreated water [5].

Infections with STH are mainly caused by ingestion of environ-mentally resistant eggs from contaminated soil (*A. lumbricoides* and *T. trichiura*) or active penetration of the skin by larvae after maturation in the soil (hookworms) [5]. The World Health Organization (WHO) set a goal of controlling morbidity due to STH by providing preventive chemotherapy to SAC and other highly vulnerable groups through mass drug administration (MDA) programs as the preferred strategy to overcome the burden of these infections [1]. The main aim was to reduce morbidity and to eliminate STH as they are a public health problem in many endemic countries [1]. Nevertheless, interventions targeting improvement of access to safe water, sanitation and hygiene (WASH) are encouraged as a long-term and sustainable control measure [6].

In Kenya, STH are endemic in 66 sub-counties that were identified based on historical data and predictive maps [7–9]. The maps were predicting the extend of STH transmission limits in the country, which resulted to the establishment of the school-based deworming programme [10]. The common anthelmintic drugs used against these infections listed by WHO are albendazole, levamisole, mebendazole and pyrantel, while ivermectin has also been registered for use against *S. stercoralis* in humans [11]. The use of these drugs is not limited to treatment of symptomatic STH infections, but also for large-scale prevention of morbidity through mass drug administration (MDA) with children living in endemic areas being the targeted population, like is the case in Kenya [10,12,13]. However, such programmes may require evaluation to determine their effectiveness in a public health setting for improved control in different geographical and environmental settings.

The National School-Based Deworming Programme (NSBDP) in Kenya was launched in 2012 by both Ministry of Health (MoH) and Ministry of Education (MoE) and has been offering MDA to 28 endemic counties in Kenya. The programme had been targeting all SAC, both enrolled and non-enrolled, as well as those in the pre-schools who live in sub-counties identified as having a high prevalence of STH (above 20%) [14]. The anthelmintic drugs are delivered by trained primary school teachers as recommended by WHO [15]. Results of the monitoring and evaluation (M&E) of the NSBDP has shown marked decline in STH infection in many of the counties involved. Approximately over five million school children were reached with over 75% coverage in seven rounds of MDA that took place during 2012–2018 and the country-wide prevalence reduced from 32.3% to 12.9% for STH, but varied prevalence rates observed in different counties [10,13]. However, there are a few counties where the burden of STH has largely remained unchanged after the treatment period [10,13]. Specifically, the three surveyed counties of Narok, Vihiga and Kisii had an initial any STH prevalence of 53.0%, 50.0% and 46.8% [16], which marginally reduced to 24.5%, 30.7% and 21.6% respectively, after the seven rounds of MDA [17]. Indicating a slow decline in prevalence of STH infection in the three counties and the schools selected for purposes of this survey (Table 1). It is expected that after seven years of annual MDA, the disease prevalence will be lowered to the point at which a reduction in the frequency of subsequent rounds of MDA can be considered. According to the WHO STH decision tree [18], the MDA regimen should be continued according to a set of endemicity classes defined by prevalence thresholds as follows: suspend MDA if STH prevalence is less than 2%; conduct MDA every 2 years if STH prevalence is between 2% and 10%; annually if between 10% and 20%; twice yearly if between 20% and 50%; and thrice yearly if greater than 50% [19]. Hence, there is need to understand the underlying factors that led to the observed slow decline of STH infection after seven rounds of annual MDA (2012 to 2018) in these three counties.

**Table 1. List of selected schools and their prevalence from 2012–2019.**

| School name | 2012 | 2013 | 2014 | 2016 | 2017 | 2018 | 2019 |
|---|---|---|---|---|---|---|---|
|  | Y1 | Y2 | Y3 | Y4 | Y5 | Y6 | Y7 |
| Karda | 60.2% | 51.4% | 29.5% | 52.8% | 49.1% | 32.4% | 28.9% |
| Nkarano | 65.7% | NS | 46.3% | NS | 63.5% | 28.8% | 38.8% |
| Ebusakami | 52.8% | 36.8% | 47.1% | 57.1% | 67.6% | 39.2% | 34.5% |
| Ebwiranyi | 65.7% | NS | 61.6% | NS | 65.7% | 49.5% | 47.7% |
| Nyambogo | 42.2% | 19.4% | 25.9% | 50% | 24.3% | 39.8% | 19.1% |
| Ameagara | 40.7% | 27.1% | 28.7% | 24.3% | 28.7% | 23.2% | 9.2% |

The WHO's Global Strategy on WASH (2015–2020) which has now been succeeded by a renewed strategy (2021–2030), strongly advocates for collaborations between WASH and NTD sectors to improve access among populations at highest risk of diseases of poverty [20,21]. The strategy aims at strengthening programmatic responses to NTDs specifically to encourage cross-sector actions in NTD control, elimination and eradication [20]. The road map includes a cross-cutting target on achieving universal access to basic water supply, sanitation and hygiene in areas endemic for NTDs by 2030 [20]. In addition, WHO guidelines recommend sanitation systems to be designed and managed safely to protect human health from microbial hazards caused by release of human excreta (and therefore helminth eggs) which can occur at every stage of sanitation of the service chain and consequently adverse health outcomes [22]. The guidelines on sanitation also cover the well-being and psychosocial dimensions of health (such as privacy, safety and dignity) needed to encourage and sustain use of sanitation services [22]. WASH and NTD sectors provide an opportunity for accelerating and sustaining disease control efforts which can be realized when there is collaboration between the two sectors, including raising awareness, using data for decision making, increasing the evidence base for action, and joint planning. Nevertheless, there has been a challenge of collaborating the two sectors noting that STH cannot be controlled without primary prevention strategies which include provision of safe drinking water, good personal hygiene and proper human waste disposal. Achieving universal large-scale WASH is an expensive and difficult undertaking for many countries especially those with NTDs [23]. Ultimately, interventions are needed to improve access to water supply, sanitation services and improvement of other preventive practices, noting that STH infection can result from contaminated crops, vegetables and fruits as well. Hence, development of appropriate WASH targets and indicators is an important component of control and elimination of STH and other NTDs [23]. WASH and environmental risk factors like poor sanitation, open defecation, low personal hygiene, lack of hand washing facilities among school children targeted for MDA is very common. Certainly, this could be the cause for the slow decline of STH infection after several rounds of MDAs. This study tested the hypothesis that there were no factors associated with slow decline in prevalence and intensity of STH among school children who had participated in the school deworming programme. Hence, the current study sought to determine the factors associated with the slow decline in STH prevalence and intensity after seven rounds of consecutive MDA in three selected endemic counties in Kenya.

## Methods

### Ethics statement

Prior to data collection, study protocol was reviewed by Scientific and Ethics Review Unit (SERU # 3775), of Kenya Medical Research Institute (KEMRI). Permission to carry out the

study was sought both at County level and school level from the headteacher or the deputy head teacher to conduct the study. While a written informed consent was sought from the study participant's parents or guardians to participate in the study. Study participants who were 13–16 years of age gave assent also to participate in the study.

## Study area

The study was conducted in March 2019 in rural settings of three STH endemic counties of Kenya namely, Narok, Kisii and Vihiga. In each selected county, one sub-county was selected with two schools being targeted. Narok County is situated in the southern part of the Great Rift Valley. Narok lies between latitudes 0˚ 50′ and 1˚ 50′ South and longitude 35˚ 28′ and 36˚ 25′ East. The county borders the Republic of Tanzania and six other counties. The population of the county is projected at around 1,157,873 persons as of 2019, with the dominant ethnic groups being the Maasai and Kalenjin [24]. The county is rich in terms of nature with the world-famous Maasai Mara National Reserve, as well as Mau Forest located within its borders. The main cash crop grown in Narok County is wheat, and large population practice pastoralism. Kisii County is among the counties that border Narok County with an estimated population of 1,266,860 people and covers an area of 1,317.5 km$^2$ [25]. The county is inhabited mostly by the Gusii ethnic group with tea farming as the main cash crop. Vihiga County is in the Western region of Kenya, whose headquarters is in Mbale. It is one of the four Counties in the former Western Province. The county boarders Nandi to the East, Kisumu County to the South, Siaya County to the West and Kakamega County to the North. The county has a population of 590,013 and an area of 563 km [24]. The predominant ethnic group is the Luhya community and the main economic activity is subsistence farming.

## Study design and setting

Using quantitative and qualitative methods, we conducted a cross-sectional study in rural settings of the three endemic counties of Kenya. In each county, two schools which had been participating in the NSBDP since 2012 were purposively sampled based on high STH prevalence levels as per the NSBDP monitoring and evaluation report of 2017 [26]. In each school, children were randomly sampled to participate in the study. Each study participant was requested to provide morning stool for parasitological examination of STH infection. In addition, an interview was conducted with each study participant using a questionnaire to capture information on; socio-demographics (child information), deworming in the previous year, household wealth information, availability and use of WASH facilities both in school and home. The interviews were conducted in Swahili for better understanding by the study participants. Community meetings with parents/guardians of the selected school children were held at each school prior to the survey to explain the purpose, procedures, potential risks, and benefits of the study. Written informed consent was obtained from parents or legal guardians of all sampled participants. In addition, participants above 13 years of age were invited to give their assent. The study profile is elaborated in Fig 1.

## Sample size determination and sampling procedure

The sample size calculation was done to estimate the number of study participants to be selected in each school. The unit of analysis was prevalence of infection among school children within the sampled schools. The sample frame was all participating primary schools in the M&E arm of the NSBDP. Three counties which had showed low reduction in prevalence (i.e. below 50%) and/or maintained high prevalence (i.e. above 30%) of infection as per the M&E report of 2017, were purposively selected [26]. Within each county, the first two schools with

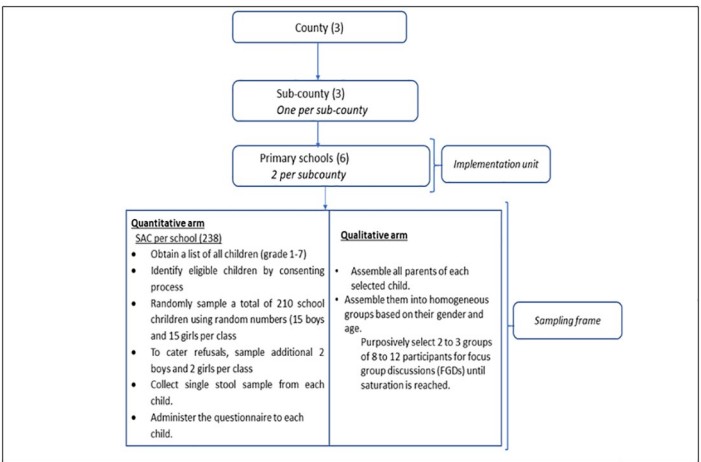

**Fig 1. Study profile.**

high reported STH infection prevalence as at 2017 M & E report were purposively selected [16]. At each selected school, the number of children to be included in the study was calculated as shown below:

$$n = \frac{pqZ_{\alpha/2}^2}{(e)^2} = \frac{0.3 \, x(1-0.3)(1.96)^2}{(0.06)^2} = 224 \, children \, per \, school$$

Accounting for refusals and non-responses, the minimum sample was increased by 6.4% to bring the total sample size to 238 children per school. Therefore, 17 boys and 17 girls aged 5–18 years from classes 1 to 7 were randomly sampled using random generated numbers. In schools where the desired sample size was not achieved because of low enrolment, all the students in classes 1 to 7 were recruited.

The study team visited each selected school to seek authority from the administrators to conduct the survey, by informing parents/ guardians of the school children. Once the recruitment had been done, each parent/guardian of the selected children were expected to accompany their child to the school for consenting purposes. Each consented child was expected to provide stool sample for STH examination and administered a questionnaire and then exist the survey.

## Sample collection

All the study enrolled children were provided with poly pots (stool containers) a day before the survey and requested to give their own fresh stool sample on collection day. The research team with the assistance of the school health teacher guided the students on stool sample collection. A trained field assistant registered all the students who provided their stool samples and assigned each of them a unique number. The name, sex, age and class of each child were recorded. The specimens collected were transported to the nearest sub-county hospital in each selected county for laboratory examination (Narok-Kilgoris, Kisii-Masaba South and Vihiga-Luanda).

## Sample processing using Kato-Katz technique

Screening for STH eggs was based on duplicate thick smears consisting of 41.7mg of stool prepared using the Kato-Katz technique [27]. The slides were examined under a microscope within 30 minutes and the counts expressed as eggs per gram (epg). The parasitological procedure was carried out by three trained medical laboratory technologists from Ministry of Health (MOH). Any resulting discrepancies in the slide readings were resolved by a senior technologist. For quality assurance purposes, 10% of the slides were re-read by a senior technologist.

## Administration of questionnaire

A structured pre-tested questionnaire developed in English and translated to Swahili language was administered to all the school children recruited for the study. The pre-testing of the questionnaire was done with school children in a separate primary school with similar settings to the selected schools in study area. Questionnaires were administered to collect data on socio-demographic, economic, and WASH factors both at household and school-levels. The socio-demographic indicators included the age, and gender of the study participant. Other questions asked were wealth information which was captured by household items owned by the family, education level of the household head or guardian and house type. In addition, participant's deworming history was also captured to know if they have been participating in the NSBDP. Finally, study participants were interviewed on their school and home WASH information. This included, type of water source, availability of a pit latrine/toilet, use of toilet and hand-washing facilities with water and soap near the toilet and shoe-wearing.

## Focus group discussions

Qualitative data was collected using a guide developed to conduct focus group discussion (FGD) with parents/guardians of the school children. The purpose of the FGDs was to collect information on study participant's perceptions, experiences and opinions of the NSBDP. Fifteen (15) FGDs were conducted in the three sub counties. Study participants for the FGD discussion were purposively selected based on their age, level of education, gender and having a child enrolled in the selected schools. Saturation model was used to determine the number of FGDs to be conducted in each study area [28]. Data was collected in the local language and in Swahili were necessary. Research assistants who were trained on the study procedures, research ethics, recruitment process and how to conduct the interviews, moderated the FGDs and note taking. The FGDs were audio recorded using an audio recorder. The interviews were conducted using either the respondent's primary language or Kiswahili or English, as appropriate. Data was transcribed verbatim and translated to English for analysis.

## Ethical statement

Ethical clearance was received from the Kenya Medical Research Institute (KEMRI), Scientific and Ethics Review Unit (SERU No. 3577). Written informed consent was sought from parents/guardians of school children while assent was sought from all the participating children in the quantitative arm of the study. In addition, written informed consent was sought from study participants of the qualitative arm of the study with specific mention of the audio recording process for the focus group discussions (FGDs). During data capture and transcription, participant names were replaced with alphanumeric unique identifiers to ensure anonymity and confidentiality.

### Data management and statistical analysis (quantitative data)

Questionnaire and laboratory data were captured electronically using the Open Data Kit (ODK) system and imported into STATA version 15.1 (STATA Corporation, College Station, TX, USA) for cleaning, management and analysis. The prevalence and mean intensity of helminth infections were calculated, and the 95% confidence interval (CIs) obtained using binomial regression and negative binomial regression models respectively, taking into account clustering by schools. Intensity of types of infection was obtained by multiplying egg counts by a factor of 24 to obtain epg and categorized as light, moderate and heavy intensity according to WHO guidelines [29]. Factors associated with STH prevalence were assessed first using univariable analysis and described as odds ratio (OR) using logistic regression model at two levels; pupils nested within schools selected within counties. For multivariable analysis, significant variables (p-value < 0.05) were selected and included in the model in a sequential (block-wise) variable selection method which selected covariates meeting the set criterion. Adjusted OR (aOR), of the most parsimonious model, were obtained by mutually adjusting all the minimum generated variables using multivariable logistic regression model at two levels; pupils nested within schools selected within counties.

### Data management and analysis (qualitative data)

Qualitative data were collected through voice records using interview guides. The voice records were transcribed and back translated by the study team. The codes were entered to QSR NVIVO version 12 for management and analysis. A code sheet was created using the FGD guide to code the data using textual data to create a master sheet analysis giving all the responses a specific theme. Thematic analysis was used where responses were categorized into themes and then ideas formulated by looking at the patterns of responses. The analyzed data were presented in text format. Representative quotes were embedded within the results to illustrate themes, with minor grammatical alterations to improve readability. Some of the predetermined themes were WASH challenges in schools and homes, treatment of school children only and leaving out the rest of the community members from the annual MDA.

## Results

### Socio-demographics, school and household information

A total of 1,874 school children from six primary schools were surveyed across three counties: 551 (30.3%) children from Narok, 756 (40.3%) from Vihiga, and 567 (30.3%) from Kisii (Table 2). In terms of age categories majority (1,800) of the participants were aged between 5 to 14 years (95.9%) and an equal gender representation of 939 (50.1%) boys and 935 (49.9%) girls (Table 2).

### Socio-demographic characteristics of FGDs respondents

In qualitative analysis, 15 FGDs were conducted with 141 parents/ guardians of school children 57.5% being female and 42.5% being male. Majority of the study participants had primary level of education (55.3%) while farming was the main occupation of the study participants (73.8%) (Table 3). The qualitative results are presented with the relevant verbatim quotes according to the four thematic areas that emerged from the data.

### FGDs code book

**NK-KR-MALE FGD:** Narok-Nkarano- male FGD**; VH-EW-FEMALE FGD:** Vihiga-Ebwiranyi- female FGD; **NK-NR-MALE FGD:** Narok-Nkarano- male FGD; **KS-NS-FEMALE FGD:**

**Table 2. Overall prevalence of STH infections among the surveyed children.**

| Factors | No. of children sampled n (%) | Prevalence of infections %(95%CI); n | | | | Mean intensity of infection epg(95%CI) | | | |
|---|---|---|---|---|---|---|---|---|---|
| | | Any STH infection | *A. lumbricoides* | Hookworm | *T. trichiura* | Any STH infection | *A. lumbricoides* | Hookworm | *T. trichiura* |
| **Overall** | **1,874 (100.0%)** | **30.8% (28.7–32.9)** | **24.9% (23.1–27.0)** | **0.3% (0.1–0.6)** | **11.4% (10.1–12.9)** | **3875 (3029–4955)** | **3771 (2847–4995)** | **5 (0–77)** | **96 (65–142)** |
| *Counties* | | | | | | | | | |
| **Narok** | 567 (30.3%) | 34.3% (30.6–38.5) | 21.4% (18.2–25.1) | 0.4% (0.0–1.5) | 19.3% (16.3–22.9) | 3662 (2403–5581) | 3423 (1935–6057) | 4 (0–270) | 235 (136–404) |
| **Vihiga** | 551 (29.4%) | 40.7% (37.4–44.4) | 36.0% (32.8–39.6) | 0.4% (0.1–1.2) | 13.9% (11.7–16.7) | 5471 (3985–7511) | 5391 (3817–7614) | 9 (0–338) | 63 (37–106) |
| **Kisii** | 756 (40.3%) | 13.8% (11.2–17.0) | 13.6% (11.1–16.8) | 0 | 0.2% (0.0–1.3) | 1931 (932–4003) | 1927 (924–4015) | 0 | 5 (0–2865) |
| *Schools* | | | | | | | | | |
| **Karda** | 249 (13.3%) | 28.9% (23.8–35.1) | 8.8% (5.9–13.2) | 0 | 24.9% (20.1–30.9) | 1946 (968–3912) | 1548 (379–6330) | 0 | 398 (196–807) |
| **Nkarano** | 302 (16.1%) | 38.8% (33.6–44.7) | 31.8% (26.9–37.5) | 0.7% (0.2–2.7) | 14.7% (11.2–19.3) | 5091 (3032–8549) | 4985 (2748–9041) | 7 (0–492) | 99 (44–226) |
| **Ebusakami** | 397 (21.2%) | 34.5% (30.1–39.5) | 29.5% (25.3–34.3) | 0.8% (0.2–2.3) | 10.8% (8.2–14.4) | 5249 (3199–8612) | 5150 (2971–8928) | 18 (0–638) | 81 (34–189) |
| **Ebwiranyi** | 359 (19.2%) | 47.7% (42.8–53.2) | 43.4% (38.5–48.9) | 0 | 17.4% (13.9–21.8) | 5720 (3817–8571) | 5661 (3669–8733) | 0 | 43 (23–81) |
| **Nyambogo** | 268 (14.3%) | 19.1% (14.9–24.5) | 19.1% (14.9–24.5) | 0 | 0 | 2945 (1215–7138) | 2945 (1215–7138) | 0 | 0 |
| **Ameagara** | 199 (15.9%) | 9.2% (6.4–13.1) | 8.8% (6.1–12.7) | 0 | 0.3% (0.0–2.4) | 1032 (297–3582) | 1022 (286–3646) | 0 | 10 (0–5385) |
| *Age categories* | | | | | | | | | |
| **5–14 years** | 1,800 (95.9%) | 31.3% (29.2–33.5) | 25.5% (23.6–27.7) | 0.3% (0.1–0.7); | 11.6% (10.2–13.1) | 3997 (3117–5126) | 3891 (2932–5164) | 5 (0–81) | 98 (66–146) |
| **≥ 15 years** | 74 (3.9%) | 20.3% (12.9–31.8) | 12.2% (6.6–22.4) | 0 | 8.1% (3.8–17.5) | 1033 (220–4849) | 983 (120–8035) | 0 | 50 (5–488) |
| *Gender* | | | | | | | | | |
| **Male** | 939 (50.1%) | 30.4% (27.5–33.5) | 24.7% (22.0–27.6) | 0.2% (0.0–0.9) | 11.3% (9.4–13.5) | 3717 (2618–5276) | 3591 (2406–5359) | 0 (0–2) | 126 (72–220) |
| **Female** | 935 (49.9%) | 31.2% (28.3–34.3) | 25.3% (22.7–28.3) | 0.3% (0.1–1.0) | 11.5% (9.6–13.8) | 4033 (2855–5696) | 3952 (2663–5864) | 10 (0–377) | 67 (39–114) |

Kisii-Nyamengesa-female FGD; **KS-NB-FEMALE FGD: Kisii-** Nyambogo-female FGD; **KS-EB-MALE FGD:** Kisii- Ebachwa- male FGD; **NK-KR-MALE FGD:** Narok-Karda- Male FGD; **NK-KR-FEMALE FGD:** Narok- Karda- female FGD.

## Prevalence and mean intensity of STH infections

The overall prevalence of any STH infection was 30.8% (95%CI: 28.7–32.9) and mean intensity of 3875 epg (95%CI: 3029–4955) with species specific intensity of 3771 epg (95%CI: 2847–4995) for *A. lumbriocoides*, *5* epg (95%CI: 0–77*)* for hookworm and 96 epg (95%CI: 65–142) for *T. trichiura* infections (Table 2). For any STH infection, the prevalence was highest in Vihiga County at 40.7% (95%CI: 37.4–44.4), followed by Narok at 34.3% (95%CI: 30.6–38.5) and least in Kisii at 13.8% (95%CI: 11.2–17.0) (Table 2). Among the schools, STH prevalence was highest in Ebwiranyi School at 47.7% (95%CI: 42.8–53.2), followed by Nkarano at 38.8% (95%CI: 33.6–44.7), Ebusakami at 34.5% (95%CI: 30.1–39.5), and the least in Nyambogo at 9.2% (95%CI: 6.4–13.1) (Fig 2). While the intensity was highest in Ebwiranyi School at 5720

**Table 3. Socio-demographic characteristics of FGDs respondents.**

| Description | Freq (N = 141) | Percent (%) |
|---|---|---|
| Gender | | |
| Female | 81 | 57.5 |
| Male | 60 | 42.5 |
| Age in years | | |
| 17–25 years | 14 | 9.9 |
| 26–34 years | 42 | 29.8 |
| 35–43 years | 41 | 29.1 |
| 44–52 years | 25 | 17.7 |
| 53–61 years | 14 | 9.9 |
| 62 and above | 5 | 3.6 |
| Level of education | | |
| Primary level | 78 | 55.3 |
| Secondary level | 53 | 37.6 |
| Post-secondary/college | 1 | 0.7 |
| No education | 9 | 6.4 |
| Occupation | | |
| Business | 14 | 9.9 |
| skilled laborer | 5 | 3.6 |
| other | 2 | 1.4 |
| farmer | 104 | 73.8 |
| Housewife/no job | 8 | 5.7 |
| casual laborer | 8 | 5.7 |

epg (95%CI: 3817–8571), followed by Ebusakami at 5249 epg (95%CI: 3199–8612), and least in Ameagara at 1032 epg (95%CI: 297–3582) respectively (Fig 2).

## Univariable and multivariable analysis of factors associated with infection

For each type of infection (any STH species, *A. lumbriocoides*, hookworm and *T. trichiura*), a univariable model was used to assess socio-demographics, family information, household and school level WASH factors and significant associations (OR) are shown in Table 3. Factors that met the inclusion criteria ($p < 0.05$) were then fitted in a multivariable model and adjusted until the most parsimonious models were attained. The factors that remained significant (aOR) after adjusting the models are shown for each infection in Table 4. Significant associations with increased odds of any STH infection were shown in children from Narok and Vihiga (aOR = 3.29 (95%CI: 1.79–6.05); $p < 0.001$) and (aOR = 3.78 (95%CI: 1.81–7.88); $p < 0.001$) respectively. Those who reported not washing their hands with soap and water at handwashing place the last time they defecated were also significantly associated with any STH infection (aOR = 1.91 (95%CI: 1.13–3.20); $p = 0.015$) (Table 5).

This was echoed by the parents/guardians of the SAC as per the themes below;

## Poor WASH factors both in school and at home

Poor WASH factors both in school and at home were mentioned as contributing factors to STH infection. Parents/guardians of the school children who were selected to participate in the FGDs shared their experiences on the NSBDP. It was agreed that there were major benefits achieved because of the programme. Notwithstanding, there were challenges that might have

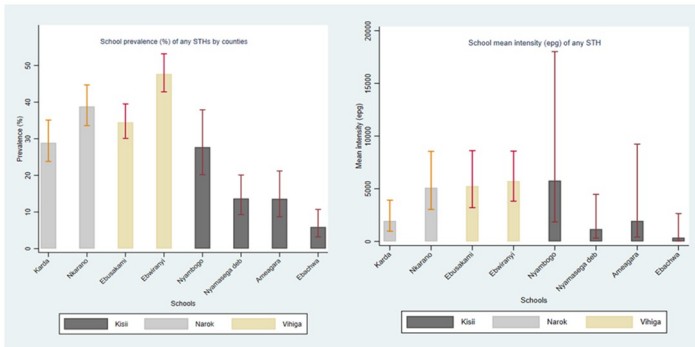

**Fig 2. Showing school level prevalence (panel A) and mean intensity (panel B) of STH infections**

contributed to the slow decline of STH infection over the years even after receiving several rounds of MDA. One of the challenges mentioned by the study participants was poor WASH conditions both at school and home. Most of the study participants felt that the NSBDP was giving the deworming drugs to school children while the sanitary facilities were in deplorable condition which can hinder the control of the parasitic diseases. In other cases, the toilets were clean but few, not according to the recommended ratio by the MoE. In most schools there were no leaky tins for hand washing after visiting the toilet. Below are some selected extracts from the transcripts:

> **P6:** *"If you look at the school environment it is not very good. Nowadays they have separate toilets for boys and girls. But if you go there, you will find that the toilets are very dirty, very dirty. So, the cleanliness around the school is not very good"*
>
> **NK-KR-MALE FGD**.

> **P8:** *"We are not satisfied with the toilets, they are very few and the children are many, secondly the drainage of those toilets are poor"*
>
> **KS-NS-FEMALE FGD**.

> **P4:** *"In my opinion I can say those toilets have no leaky tins. It is good to have them so that the children can wash their hands and the teachers should train them"*
>
> **KS-NS-FEMALE FGD**.

In addition, most school children were not practising good personal hygiene and were walking barefooted probably due to low socio-economic status in this setting which could also be exposing them to STH infection. Also, not owning and using a pit latrine at home due to various reasons like poverty, lack of awareness to construct and maintain basic toilets would have been a contributing factor to the slow decline of worms due to open defecation and environmental contamination with fecal material. Examples of extracts from the transcripts are given below:

> **P2**: *"I don't know whether you have walked to see the toilet for girls in this school, they are dirty, there is urine all over. Other children have got no shoes, they walk bare footed, so they must get worms"*
>
> **VH-EW-FEMALE FGD**.

**Table 4. Univariable analysis of socio-demographic, household and school level risk factors associated with infections among school children.**

| Factors | No. of children sampled n/N (%) | Univariable logistic [OR (95%CI); p-value] | | | |
| --- | --- | --- | --- | --- | --- |
| | | Any STH infection n = 571 | *A. lumbricoides* n = 464 | Hookworm n = 5 | *T. trichiura* n = 212 |
| Overall prevalence | **1,874 (100.0)** | **30.77 (28.74–32.94)** | **24.99 (23.09–27.04)** | **0.27 (0.11–0.65)** | **11.41 (10.05–12.95)** |
| DEMOGRAPHICS | | | | | |
| County | | | | | |
| Kisii | 567 (30.3) | **Reference** | | | |
| Narok | 551 (29.4) | **3.25(2.41–4.38); p<0.001*** | **1.72(1.25–2.36); p = 0.001*** | 0.91(0.15–5.49); p = 0.922 | **133.34(18.54–959.14);p<0.001*** |
| Vihiga | 756 (40.3) | **4.28(3.23–5.67); p<0.001*** | **3.56(2.68–4.74); p<0.001*** | - | **90.09(12.53–647.66); p<0.001*** |
| Age | | | | | |
| 5–14 years | 1,800 (95.9) | **1.78(1.00–3.17); p = 0.048*** | **2.48(1.22–5.02); p = 0.012*** | - | 1.48(0.63–3.45); p = 0.363 |
| ≥ 15 years | 74 (3.9) | **Reference** | | | |
| Gender | | | | | |
| Male | 939 (50.1) | **Reference** | | | |
| Female | 935 (49.9) | 1.03(0.85–1.26); p = 0.702 | 1.04(0.84–1.28); p = 0.738 | 1.51(0.25–9.04); p = 0.654 | 1.02(0.77–1.36); p = 0.871 |
| Family information | | | | | |
| Number of household members | | | | | |
| ≤ 5 | 589 (31.4) | **Reference** | | | |
| 5–10 | 1,184 (63.2) | 1.12(0.90–1.39); p = 0.293 | 1.06(0.84–1.33); p = 0.647 | 1.99(0.22–17.89); p = 0.537 | **1.53(1.09–2.14); p = 0.014*** |
| >10 | 101 (5.4) | 1.09(0.69–1.73); p = 0.701 | 1.04(0.64–1.69); p = 0.883 | - | **1.88(1.01–3.51); p = 0.045*** |
| Number of household siblings | | | | | |
| ≤ 5 | 1,437 (76.7) | **Reference** | | | |
| 6–10 | 413 (22.0) | 1.08(0.85–1.37); p = 0.496 | 0.98(0.76–1.26); p = 0.847 | - | **1.76(1.28–2.42); p<0.001*** |
| >10 | 24 (1.3) | 1.38(0.59–3.18); p = 0.448 | 1.50(0.64–3.54); p = 0.353 | - | 1.82(0.61–5.39); p = 0.282 |
| No. of siblings attending school | | | | | |
| ≤ 5 | 1,714 (91.5) | **Reference** | | | |
| >5 | 160 (8.6) | 1.12(0.79–1.59); p = 0.499 | 0.93(0.64–1.36); p = 0.706 | - | **1.58(1.01–2.47); p = 0.046*** |
| No. of siblings under five | | | | | |
| None | 1,203 (64.2) | **Reference** | | | |
| 1–3 | 667 (35.6) | 1.14(0.93–1.40); p = 0.186 | 1.15(0.93–1.43); p = 0.200 | 0.45(0.05–4.03); p = 0.476 | 0.95(0.70–1.28); p = 0.734 |
| >3 | 4 (0.2) | 7.13(0.73–68.78); p = 0.089 | 3.17(0.44–22.62); p = 0.250 | - | 2.55(0.26–24.67); p = 0.419 |
| Household wealth information and possessions | | | | | |
| Level of education of HH | | | | | |
| No education | 145 (7.7) | **Reference** | | | |
| Primary education | 592 (31.6) | 0.92(0.61–1.37); p = 0.682 | 0.87(0.56–1.35); p = 0.532 | - | 1.01(0.56–1.80); p = 0.976 |
| Secondary and above | 426 (22.7) | 0.99(0.65–1.49); p = 0.945 | 1.07(0.68–1.68); p = 0.757 | 1.80(0.30–10.82); p = 0.520 | 0.92(0.50–1.69); p = 0.792 |
| Don't know | 711 (37.9) | 1.35(0.91–2.01); p = 0.132 | 1.35(0.89–2.07); p = 0.159 | - | 1.08(0.61–1.90); p = 0.796 |
| Occupation of HH | | | | | |

*(Continued)*

**Table 4.** (Continued)

| Factors | No. of children sampled n/N (%) | Univariable logistic [OR (95%CI); p-value] | | | |
|---|---|---|---|---|---|
| | | Any STH infection n = 571 | A. lumbricoides n = 464 | Hookworm n = 5 | T. trichiura n = 212 |
| Farming | 617 (32.9) | **Reference** | | | |
| Small business | 502 (26.8) | **1.53(1.18–1.98); p = 0.001***  | **1.59(1.22–2.10); p = 0.001***  | - | **1.55(1.07–2.24); p = 0.020***  |
| Employed | 268 (14.3) | 0.97(0.69–1.3); p = 0.850 | 0.99(0.70–1.42); p = 0.988 | - | 1.07(0.66–1.73); p = 0.787 |
| Others | 487 (25.9) | **1.31(1.01–1.70); p = 0.042***  | **1.40(1.06–1.86); p = 0.018***  | 0.84(0.14–5.05); p = 0.849 | 1.27(0.86–1.87); p = 0.224 |
| **Type of wall** | | | | | |
| Stone/bricks/cemented | 468 (24.9) | **Reference** | | | |
| Clay/mud | 1,381 (73.7) | **1.28(1.02–1.62); p = 0.037***  | 1.28(0.99–1.65); p = 0.051 | - | 1.11(0.79–1.56); p = 0.526 |
| Wood | 8 (0.4) | 0.89(0.18–4.50); p = 0.895 | 1.20(0.24–6.04); p = 0.824 | - | - |
| Iron sheets | 16 (0.9) | 0.38(0.09–1.72); p = 0.210 | 0.51(0.12–2.30); p = 0.385 | - | - |
| Other | 1 (0.1) | - | - | - | - |
| **Type of floor** | | | | | |
| Cement/tiles/linoleum | 554 (29.6) | **Reference** | | | |
| Wooden | 2 (0.1) | 2.37(0.15–38.19); p = 0.542 | 3.29(0.20–53.08); p = 0.400 | - | - |
| Earth/sand | 1,318 (70.3) | 1.11(0.87–1.34); p = 0.502 | 1.14(0.90–1.44); p = 0.276 | 1.69(0.19–15.14); p = 0.640 | 0.90(0.66–1.23); p = 0.505 |
| **Type of roof** | | | | | |
| Tiles | 4 (0.2) | **Reference** | | | |
| Iron sheets | 1,816 (96.9) | 1.34(0.14–12.89); p = 0.801 | 1.00(0.10–9.67); p = 0.997 | - | 0.39(0.04–3.72); p = 0.410 |
| Grass/thatch | 52 (2.8) | 1.25(0.12–13.00); p = 0.852 | 0.90(0.09–9.46); p = 0.930 | - | 0.39(0.03–4.39); p = 0.447 |
| Other | 2 (0.1) | - | - | - | - |
| **Owns car** | | | | | |
| Yes | 82 (4.4) | **Reference** | | | |
| No | 1,792 (95.6) | 1.13(0.68–1.84); p = 0.637 | 1.09(0.65–1.84); p = 0.745 | - | 0.66(0.36–1.22); p = 0.182 |
| **Owns motorbike** | | | | | |
| Yes | 374 (19.9) | **Reference** | | | |
| No | 1,500 (80.0) | 0.95(0.74–1.21); p = 0.655 | 0.95(0.73–1.23); p = 0.683 | - | 0.86(0.61–1.22); p = 0.407 |
| **Owns bicycle** | | | | | |
| Yes | 298 (15.9) | **Reference** | | | |
| No | 1,576 (84.1) | 0.88(0.67–1.15); p = 0.341 | 0.85(0.64–1.12); p = 0.255 | 0.76(0.08–6.80); p = 0.804 | 1.48(0.96–2.29); p = 0.078 |
| **Owns mobile phone** | | | | | |
| Yes | 1,799 (96.0) | **Reference** | | | |
| No | 75 (4.0) | 0.95(0.57–1.58); p = 0.844 | 1.04(0.61–1.77); p = 0.889 | 6.09(0.67–55.19); p = 0.108 | 0.80(0.36–1.78); p = 0.591 |
| **Owns radio** | | | | | |
| Yes | 1,582 (84.4) | **Reference** | | | |
| No | 292 (15.6) | 1.18(0.90–1.54); p = 0.224 | 1.19(0.90–1.59); p = 0.208 | 1.35(0.15–12.14); p = 0.788 | **1.49(1.04–2.13); p = 0.029***  |

(Continued)

**Table 4.** (Continued)

| Factors | No. of children sampled n/N (%) | Univariable logistic [OR (95%CI); p-value] | | | |
|---|---|---|---|---|---|
| | | Any STH infection n = 571 | A. lumbricoides n = 464 | Hookworm n = 5 | T. trichiura n = 212 |
| **Owns television** | | | | | |
| Yes | 617 (32.9) | Reference | | | |
| No | 1,257 (67.1) | 1.01(0.82–1.25); p = 0.924 | 1.10(0.87–1.38); p = 0.398 | - | 0.77(0.58–1.04); p = 0.090 |
| **Owns sofa set** | | | | | |
| Yes | 1,039 (55.4) | Reference | | | |
| No | 835 (44.6) | 0.85(0.69–1.03); p = 0.103 | 0.98(0.79–1.21); p = 0.860 | 1.87(0.31–11.23); p = 0.493 | **0.61(0.45–0.82); p = 0.001*** |
| **Owns electricity** | | | | | |
| Yes | 719 (38.4) | Reference | | | |
| No | 1,155 (61.6) | **1.41(1.15–1.74); p = 0.001*** | **1.75(1.39–2.21); p<0.001*** | 0.94(0.15–5.62); p = 0.943 | 0.83(0.62–1.11); p = 0.201 |
| **TREATMENT FACTORS** | | | | | |
| **Treated for worms in the last year** | | | | | |
| Yes | 1,079 (57.6) | Reference | | | |
| No | 768 (40.9) | **1.38(1.14–1.69); p = 0.001*** | **1.48(1.19–1.84); p<0.001*** | 0.94(0.16–5.62); p = 0.943 | **1.37(1.02–1.83); p = 0.031*** |
| Don't know | 27 (1.4) | 1.29(0.58–2.92); p = 0.529 | 1.02(0.1–2.56); p = 0.969 | - | 1.56(0.53–4.60); p = 0.417 |
| **Place of treatment** | | | | | |
| School | 1,031 (95.6) | Reference | | | |
| Health center | 20 (1.9) | 0.46(0.13–1.59); p = 0.220 | 0.39(0.09–1.71); p = 0.215 | - | 0.48(0.06–3.66); p = 0.483 |
| Home | 26 (2.4) | 1.92(0.87–4.23); p = 0.106 | 1.31(0.54–3.15); p = 0.548 | - | **2.76(1.08–7.04); p = 0.033*** |
| Others | 2 (0.2) | 2.62(0.16–41.98); p = 0.497 | 3.56(0.22–57.04); p = 0.371 | - | - |
| **No. of tablets administered** | | | | | |
| 1 | 1,032 (95.6) | Reference | | | |
| 2 | 41 (3.8) | 1.07(0.54–2.12); p = 0.850 | 0.85(0.39–1.87); p = 0.691 | - | 1.25(0.48–3.26); p = 0.645 |
| 3 | 3 (0.3) | - | - | - | - |
| Not sure | 3 (0.3) | - | - | - | - |
| **Color of the tablets** | | | | | |
| White | 1,021 (97.2) | Reference | | | |
| Yellow | 10 (1.0) | **3.95(1.11–14.10); p = 0.034*** | **3.57(1.02–12.44); p = 0.046*** | - | 1.01(0.13–8.07); p = 0.991 |
| Blue | 3 (0.3) | 1.32(0.12–14.58); p = 0.823 | - | - | 4.56(0.41–50.68); p = 0.217 |
| Others | 17 (1.6) | 1.09(0.38–3.14); p = 0.863 | 0.77(0.22–2.68); p = 0.676 | - | 1.95(0.55–6.90); p = 0.300 |
| **HOUSEHOLD LEVEL FACTORS** | | | | | |
| **Presence of toilet/latrine** | | | | | |
| Yes | 1,714 (91.5) | Reference | | | |
| No | 160 (8.5) | **1.70(1.22–2.37); p = 0.002*** | 1.15(0.80–1.66); p = 0.443 | **16.19(2.69–97.64); p = 0.002*** | **2.11(1.39–3.19); p<0.001*** |
| **Toilet/latrine shared** | | | | | |
| Yes | 730 (42.6) | Reference | | | |

*(Continued)*

**Table 4.** (Continued)

| Factors | No. of children sampled n/N (%) | Univariable logistic [OR (95%CI); p-value] | | | |
|---|---|---|---|---|---|
| | | Any STH infection n = 571 | A. lumbricoides n = 464 | Hookworm n = 5 | T. trichiura n = 212 |
| No | 984 (57.4) | 0.87(0.71–1.07); p = 0.196 | 0.91(0.73–1.13); p = 0.390 | 0.74(0.05–11.83); p = 0.830 | 0.87(0.64–1.18); p = 0.384 |
| **Place for urination** | | | | | |
| In my latrine/toilet | 1,559 (83.2) | **Reference** | | | |
| In a latrine outside compound | 88 (4.7) | 1.44(0.93–2.25); p = 0.104 | 1.24(0.78–1.99); p = 0.374 | 5.89(0.61–57.27); p = 0.126 | 0.93(0.46–1.88); p = 0.832 |
| Around/outside compound (e.g. in the bush) | 182 (9.7) | 1.02(0.73–1.42); p = 0.912 | 0.78(0.54–1.13); p = 0.193 | 2.83(0.29–27.39); p = 0.368 | **1.54(1.00–2.36); p = 0.048*** |
| In the bathroom/shower | 45 (2.4) | 0.93(0.48–1.79); p = 0.831 | 0.84(0.41–1.72); p = 0.641 | - | 1.02(0.39–2.61); p = 0.973 |
| **Place for defecating** | | | | | |
| In my latrine/toilet | 1,717 (91.6) | **Reference** | | | |
| In a latrine outside compound | 60 (3.2) | 1.43(0.84–2.43); p = 0.186 | 0.90(0.49–1.67); p = 0.741 | 9.59(0.98–93.60); p = 0.052 | 1.87(0.96–3.67); p = 0.067 |
| Around/outside compound (e.g. in the bush) | 97 (5.2) | 1.21(0.78–1.85); p = 0.399 | 0.83(0.51–1.36); p = 0.462 | 5.96(0.61–57.82); p = 0.124 | **2.06(1.22–3.48); p = 0.007*** |
| **Used a latrine/toilet the last time they defecated** | | | | | |
| Yes | 1,628 (86.9) | **Reference** | | | |
| No | 246 (13.1) | 1.21(0.91–1.62); p = 0.186 | 1.09(0.81–1.49); p = 0.546 | 1.67(0.19–15.01); p = 0.647 | 1.43(0.97–2.11); p = 0.070 |
| **Usually used of wiping material after defecation** | | | | | |
| Always | 568 (30.3) | **Reference** | | | |
| Sometimes | 1,034 (55.2) | 1.05(0.84–1.32); p = 0.645 | 1.12(0.88–1.43); p = 0.347 | 1.65(0.17–15.89); p = 0.665 | 0.79(0.58–1.09); p = 0.150 |
| Never | 272 (14.5) | 1.15(0.84–1.57); p = 0.381 | 1.20(0.86–1.67); p = 0.278 | 2.09(0.13–33.65); p = 0.601 | 0.77(0.48–1.22); p = 0.262 |
| **Type of wiping material used at home** | | | | | |
| Tissue paper | 597 (31.9) | **Reference** | | | |
| Water | 4 (0.2) | 0.68(0.07–6.67); p = 0.748 | - | - | 2.13(0.22–20.76); p = 0.514 |
| Other types of paper | 577 (30.8) | 1.05(0.82–1.34); p = 0.711 | 1.14(0.88–1.48); p = 0.309 | 1.04(0.06–16.59); p = 0.981 | 0.82(0.58–1.16); p = 0.263 |
| Leaves | 658 (35.1) | **0.75(0.59–0.96); p = 0.020*** | 0.78(0.59–1.01); p = 0.062 | 2.74(0.28–26.37); p = 0.384 | **0.64(0.45–0.91); p = 0.013*** |
| Nothing | 38 (2.0) | 0.95(0.47–1.93); p = 0.896 | 0.53(0.22–1.29); p = 0.167 | - | 1.45(0.62–3.39); p = 0.398 |
| **Place (e.g. container, basin, sink) to wash hands after defecating** | | | | | |
| Yes | 841 (44.9) | **Reference** | | | |
| No | 1,033 (55.1) | 1.05(0.86–1.23); p = 0.643 | 1.03(0.83–1.27); p = 0.797 | 1.22(0.20–7.33); p = 0.825 | 1.05(0.78–1.39); p = 0.751 |
| **Water usually available at handwashing place** | | | | | |
| Always | 418 (49.7) | **Reference** | | | |
| Sometimes | 416 (49.5) | **1.53(1.13–2.06); p = 0.005*** | **1.84(1.34–2.54); p<0.001*** | - | 1.25(0.81–1.94); p = 0.305 |
| Never | 7 (0.8) | - | - | - | - |
| **Soap usually available at handwashing place** | | | | | |
| Always | 268 (31.9) | **Reference** | | | |

(Continued)

**Table 4.** (Continued)

| Factors | No. of children sampled n/N (%) | Univariable logistic [OR (95%CI); p-value] | | | |
|---|---|---|---|---|---|
| | | Any STH infection n = 571 | A. lumbricoides n = 464 | Hookworm n = 5 | T. trichiura n = 212 |
| Sometimes | 446 (53.0) | 1.03(0.74–1.44); p = 0.853 | 1.26(0.88–1.81); p = 0.199 | 0.28(0.02–4.55); p = 0.373 | 0.76(0.48–1.21); p = 0.254 |
| Never | 127 (15.1) | 1.02(0.64–1.61); p = 0.944 | 1.21(0.74–1.99); p = 0.441 | - | 0.69(0.35–1.38); p = 0.297 |
| **Washed hands with soap and water at this place the last time they defecated** | | | | | |
| Yes | 595 (70.8) | **Reference** | | | |
| No | 246 (29.3) | 0.99(0.72–1.37); p = 0.956 | 0.86(0.61–1.22); p = 0.402 | - | 1.36(0.87–2.15); p = 0.180 |
| **Usually washed hands with soap and water after defecating** | | | | | |
| Always | 224 (11.9) | **Reference** | | | |
| Sometimes | 1,288 (68.7) | 0.91(0.67–1.24); p = 0.566 | 0.93(0.67–1.29); p = 0.665 | - | 0.93(0.59–1.47); p = 0.769 |
| Never | 362 (19.3) | 1.10(0.77–1.58); p = 0.597 | 1.04(0.71–1.52); p = 0.858 | - | 1.29(0.78–2.16); p = 0.323 |
| **Main source of drinking water** | | | | | |
| Improved | 318 (16.9) | **Reference** | | | |
| Unimproved | 1,556 (83.0) | 1.26(0.96–1.65); p = 0.099 | 1.16(0.87–1.55); p = 0.302 | - | 1.51(0.9–2.3); p = 0.060 |
| **Usually used the drinking water source** | | | | | |
| Always | 1,074 (57.3) | **Reference** | | | |
| Sometimes | 776 (41.4) | 1.06(0.87–1.29); p = 0.587 | 1.06(0.86–1.31); p = 0.584 | 0.92(0.15–5.54); p = 0.931 | 1.18(0.88–1.57); p = 0.265 |
| Never | 24 (1.3) | 0.46(0.15–1.35); p = 0.156 | 0.61(0.21–1.81); p = 0.374 | - | 0.36(0.05–2.68); p = 0.318 |
| **SCHOOL LEVEL FACTORS** | | | | | |
| **Place for defecating** | | | | | |
| Latrine/toilet at school | 1,840 (98.2) | **Reference** | | | |
| Latrine near the school | 1 (0.1) | - | - | - | - |
| Around/outside school compound (e.g. in the bush) | 31 (1.7) | 0.78(0.35–1.75); p = 0.544 | 0.44(0.15–1.26); p = 0.126 | - | 1.50(0.57–3.95); p = 0.410 |
| Waited/held it until school is over | 2 (0.1) | - | - | - | - |
| **Place for urination** | | | | | |
| Latrine/toilet at school | 1,837 (98.0) | **Reference** | | | |
| To a latrine near the school | 5 (0.3) | 1.49(0.25–8.95); p = 0.662 | 1.98(0.33–11.86); p = 0.456 | - | - |
| Around/outside school compound (e.g. in the bush) | 32 (1.7) | 0.63(0.27–1.45); p = 0.277 | 0.31(0.09–1.01); p = 0.052 | - | 1.44(0.55–3.79); p = 0.456 |
| **Used a latrine/toilet the last time they defecated** | | | | | |
| Yes | 1,753 (93.5) | **Reference** | | | |
| No | 121 (6.5) | 1.25(0.84–1.84); p = 0.269 | 1.06(0.69–1.62); p = 0.782 | - | 1.40(0.83–2.37); p = 0.203 |
| **School usually provided of anal wiping material** | | | | | |
| Always | 21 (1.1) | **Reference** | | | |
| Sometimes | 85 (4.5) | 0.39(0.14–1.06); p = 0.066 | 0.64(0.20–2.03); p = 0.449 | - | **0.16(0.04–0.59); p = 0.006***|
| Never | 1,768 (94.3) | 0.59(0.25–1.43); p = 0.249 | 1.09(0.39–2.99); p = 0.868 | - | **0.32(0.12–0.84); p = 0.021***|

(*Continued*)

**Table 4.** (Continued)

| Factors | No. of children sampled n/N (%) | Univariable logistic [OR (95%CI); p-value] | | | |
|---|---|---|---|---|---|
| | | Any STH infection n = 571 | A. lumbricoides n = 464 | Hookworm n = 5 | T. trichiura n = 212 |
| **Type of wiping material used** | | | | | |
| Tissue | 159 (8.5) | **Reference** | | | |
| Water | 3 (0.2) | 1.31(0.12–14.78); p = 0.829 | - | - | 3.11(0.27–35.81); p = 0.362 |
| Other paper | 1,131 (60.4) | 1.23(0.85–1.78); p = 0.266 | **1.59(1.04–2.41); p = 0.031\*** | - | 0.79(0.48–1.27); p = 0.333 |
| Leaves | 462 (24.7) | 1.14(0.76–1.70); p = 0.518 | 1.39(0.89–2.19); p = 0.151 | - | 0.75(0.44–1.29); p = 0.299 |
| Nothing | 119 (6.4) | 0.82(0.48–1.42); p = 0.485 | 0.89(0.47–1.66); p = 0.705 | - | 0.85(0.41–1.73); p = 0.649 |
| **Place (e.g. container, basin, sink) to wash hands after defecating** | | | | | |
| Yes | 768 (40.9) | **Reference** | | | |
| No | 1,106 (59.0) | 1.06(0.86–1.29); p = 0.600 | 1.17(0.94–1.45); p = 0.152 | 0.46(0.08–2.77); p = 0.398 | 0.99(0.74–1.33); p = 0.980 |
| **Soap usually available at handwashing place** | | | | | |
| Always | 33 (4.3) | **Reference** | | | |
| Sometimes | 157 (20.4) | 1.01(0.40–2.54); p = 0.978 | 1.13(0.43–2.99); p = 0.799 | - | **0.19(0.05–0.69); p = 0.012\*** |
| Never | 578 (75.3) | 1.82(0.78–4.28); p = 0.167 | 1.45(0.59–3.59); p = 0.420 | - | 0.87(0.33–2.31); p = 0.777 |
| **Water usually available at handwashing place** | | | | | |
| Always | 186 (24.2) | **Reference** | | | |
| Sometimes | 572 (74.5) | 1.01(0.71–1.45); p = 0.940 | 1.02(0.69–1.52); p = 0.904 | - | 0.79(0.48–1.29); p = 0.346 |
| Never | 10 (1.3) | - | - | - | - |
| **Washed hands with soap and water at this place the last time they defecated** | | | | | |
| Yes | 226 (29.4) | **Reference** | | | |
| No | 542 (70.6) | **1.98(1.37–2.87); p<0.001\*** | **2.01(1.33–3.04); p = 0.001\*** | 0.83(0.07–9.18); p = 0.878 | 1.68(0.97–2.89); p = 0.063 |
| **Usually washed hands with soap and water after defecating** | | | | | |
| Always | 40 (2.1) | **Reference** | | | |
| Sometimes | 435 (23.2) | 1.14(0.52–2.47); p = 0.749 | 1.00(0.45–2.26); p = 0.995 | - | 1.15(0.34–3.93); p = 0.818 |
| Never | 1,399 (74.7) | 1.62(0.76–3.43); p = 0.211 | 1.39(0.64–3.07); p = 0.405 | - | 1.69(0.51–5.54); p = 0.387 |
| **Drinking water usually available** | | | | | |
| Always | 527 (28.1) | **Reference** | | | |
| Sometimes | 1,101 (58.8) | 0.91(0.72–1.13); p = 0.383 | 0.99(0.78–1.25); p = 0.919 | 0.72(0.12–4.31); p = 0.717 | 0.78(0.57–1.08); p = 0.131 |
| Never | 246 (13.1) | **0.66(0.47–0.93); p = 0.018\*** | **0.60(0.41–0.89); p = 0.010\*** | - | 0.71(0.43–1.16); p = 0.170 |
| **Usually used the drinking water source** | | | | | |
| Always | 621 (38.1) | **Reference** | | | |
| Sometimes | 941 (57.8) | **0.76(0.61–0.94); p = 0.012\*** | **0.74(0.59–0.93); p = 0.011\*** | - | 0.75(0.55–1.02); p = 0.067 |
| Never | 66 (4.1) | 0.68(0.39–1.19); p = 0.182 | 0.89(0.51–1.58); p = 0.700 | - | **0.09(0.01–0.69); p = 0.020\*** |

(*Continued*)

**Table 4.** (Continued)

| Factors | No. of children sampled n/N (%) | Univariable logistic [OR (95%CI); p-value] | | | |
|---|---|---|---|---|---|
| | | Any STH infection n = 571 | *A. lumbricoides* n = 464 | Hookworm n = 5 | *T. trichiura* n = 212 |
| **Hygiene and school attendance** | | | | | |
| **Number of days they missed school in the past 5 school days,** | | | | | |
| 0 | 1,538 (82.1) | **Reference** | | | |
| 1–3 | 299 (15.9) | **1.47(1.14–1.91); p = 0.003*** | **1.35(1.02–1.78); p = 0.034*** | - | **1.66(1.17–2.36); p = 0.004*** |
| >3 | 37 (1.9) | 1.47(0.75–2.89); p = 0.258 | 1.53(0.76–3.08); p = 0.233 | - | 1.34(0.52–3.49); p = 0.546 |
| **Usually ate soil or clay** | | | | | |
| Yes | 335 (17.9) | **Reference** | | | |
| No | 1,539 (82.1) | 1.29(0.99–1.67); p = 0.064 | 1.17(0.89–1.56); p = 0.266 | - | 1.49(0.98–2.26); p = 0.061 |
| **Type of shoes the child was wearing** | | | | | |
| Closed shoes | 1,443 (77.0) | **Reference** | | | |
| Sandals | 393 (20.9) | 0.86(0.67–1.10); p = 0.245 | **0.72(0.55–0.95); p = 0.020*** | 1.21(0.13–11.71); p = 0.866 | 1.19(0.85–1.67); p = 0.302 |
| No shoes | 38 (2.0) | 1.88(0.98–3.63); p = 0.059 | 1.95(1.00–3.79); p = 0.050 | **13.19(1.34–129.93); p = 0.027*** | 0.46(0.11–1.92); p = 0.284 |

*Variables significant at p<0.05 were selected for multivariable model

> **P1:** *Another thing, in this area a lot of households don't have toilets. When it rains it sweeps everything to the river and people drink the water causing stomach aches and things like that*
>
> **NK-NR-MALE FGD.**

## Multivariable analysis of factors associated with infection

Increased odds of infection with *A. lumbricoides* was associated with geographical location of Narok and Vihiga counties (OR = 2.42 (95%CI: 1.38–4.26); p< 0.0020) and (OR = 3.58 (95% CI: 1.75–7.32); p<0.001) respectively (Table 5). Also, lack of connection to electric power supply (aOR = 1.90 (95%CI: 1.19–3.02); p = 0.007) and those who reported not washing their hands with soap and water at handwashing place at school last time they defecated (OR = 2.06 (95%CI: 1.21–3.50); p = 0.008) (Table 5). Hookworm infections were low (n = 5) and therefore insufficient observations affected the convergence of the model. Nonetheless, children who reported lack of a toilet/latrine at home had increased odds of infection (OR = 14.71 (95%CI: 2.32–93.28); p = 0.004) (Table 4). Association of *T. trichiura* infection with geographical location and source of drinking water was revealed. Narok and Vihiga counties had increased odds of infection of *T. trichiura* (OR = 119.33 (95%CI:15.00–949.50); p<0.001) and (OR = 0.13 (95%CI:0.02–0.95); p = 0.044) respectively (Table 5). Not having received treatment for worms in the last year (OR = 1.55 (95%CI: 1.05–2.29); p< 0.027) and not owning household items like a sofa set (OR = 0.65 (95%CI 0.45–0.92); p< 0.016) were significantly associated with *T. trichiura* infection (Table 5). In the qualitative analysis, some of the themes that came out as possible factors contributing to slow decline of STH among school children are discussed below;

**Table 5. Multivariable analysis of risk factors associated with STH infections among children.**

| Factors | aOR (95%CI); p-value | | | |
|---|---|---|---|---|
| | Any STH (n = 571) | A.lumbricoides (n = 464) | Hookworm (n = 5) | T. trichiura (n = 212) |
| **DEMOGRAPHICS** | | | | |
| **County** | | | | |
| Kisii | Reference | Reference | - | Reference |
| Narok | 3.29 (1.79–6.05); p<**0.001**\* | 2.42 (1.38–4.26); **p = 0.002**\* | - | 119.33 (15.00–949.50); **p<0.001**\* |
| Vihiga | 3.78 (1.81–7.88); p<**0.001**\* | 3.58 (1.75–7.32); **p<0.001**\* | - | 69.97 (8.79–556.87); **p <0.001**\* |
| **Age (years)** | | | | |
| 5–14 | 2.04 (0.61–6.82);p = 0.246 | 6.30 (0.79–50.20); p = 0.082 | - | - |
| ≥15 | Reference | Reference | - | - |
| **Family information** | | | | |
| **Number of household members** | | | | |
| ≥5 | - | - | - | Reference |
| 6–10 | - | - | - | 1.32 (0.88–1.98); p = 0.173 |
| >10 | - | - | - | 1.60 (0.67–3.83); p = 0.291 |
| **Number of household siblings** | | | | |
| ≥5 | - | - | - | Reference |
| 6–10 | - | - | - | 1.35 (0.84–2.18); p = 0.216 |
| >10 | - | - | - | 1.17 (0.30–4.52); p = 0.815 |
| **Number of siblings attending school** | | | | |
| ≤ 5 | - | - | - | Reference |
| >5 | - | - | - | 0.98 (0.53–1.84); p = 0.955 |
| **Occupation** | | | | |
| Farming | Reference | Reference | - | Reference |
| Small business | 1.22 (0.71–2.08); p = 0.479 | 1.36 (0.80–2.32); p = 0.258 | - | 1.33 (0.84–2.11); p = 0.226 |
| Employed | 0.79 (0.40–1.57); p = 0.499 | 1.00 (0.49–2.03); p = 0.991 | - | 0.89 (0.50–1.56); p = 0.673 |
| Others | 0.71 (0.41–1.21); p = 0.208 | 0.75 (0.43–1.30); p = 0.308 | - | 1.28 (0.79–2.05); p = 0.312 |
| **Household wealth and possesion** | | | | |
| **Owns electricity** | | | | |
| Yes | Reference | Reference | - | - |
| No | 1.19 (0.76–1.85); p = 0.446 | 1.90 (1.19–3.02); **p = 0.007**\* | - | - |
| **Owns radio** | | | | |
| Yes | - | - | - | Reference |
| No | - | - | - | 1.43 (0.94–2.17); p = 0.092 |
| **Owns sofa set** | | | | |
| Yes | - | - | - | Reference |
| No | - | - | - | 0.65 (0.45–0.92); **p = 0.016**\* |
| **TREATMENT FACTORS** | | | | |
| **Treated for worms in the last year** | | | | |
| Yes | - | - | - | Reference |
| No | - | - | - | 1.55 (1.05–2.29); **p = 0.027**\* |
| Don't know | - | - | - | 2.86 (0.75–10.87); p = 0.123 |

(*Continued*)

**Table 5.** (Continued)

| Factors | aOR (95%CI); p-value | | | |
|---|---|---|---|---|
| | **Any STH (n = 571)** | ***A.lumbricoides* (n = 464)** | **Hookworm (n = 5)** | ***T. trichiura* (n = 212)** |
| **Tablet color administered for deworming** | | | | |
| White | Reference | Reference | - | - |
| Yellow | 4.87 (0.81–29.45); p = 0.085 | 2.66 (0.49–14.48); p = 0.256 | - | - |
| Blue | 1.36 (0.11–16.43); p = 0.811 | 1.00 | - | - |
| Others | 0.90 (0.25–3.15); p = 0.864 | 0.53 (0.11–2.55); p = 0.426 | - | - |
| **HOUSEHOLD LEVEL FACTORS** | | | | |
| **Presence of toilet at home** | | | | |
| Yes | Reference | - | Reference | Reference |
| No | 1.76 (0.94–3.30); p = 0.077 | - | **14.71 (2.32–93.28); p = 0.004*** | 1.06 (0.45–2.50); p = 0.891 |
| **Place for urination** | | | | |
| Latrine/toilet | - | - | - | Reference |
| In a latrine outside compound | - | - | - | 0.41 (0.12–1.39); p = 0.152 |
| Around/outside compound (e.g. in the bush) | - | - | - | 1.20 (0.48–2.99); p = 0.695 |
| In the bathroom/shower | - | - | - | 0.99 (0.27–3.64); p = 0.983 |
| **Place for defecating** | | | | |
| In latrine/toilet | - | - | - | Reference |
| In a latrine outside compound | - | - | - | 2.01 (0.51–7.88); p = 0.316 |
| Around/outside compound (e.g. in the bush) | - | - | - | 0.75 (0.19–2.91); p = 0.677 |
| **Type of wiping material used at home** | | | | |
| Tissue paper | Reference | - | - | Reference |
| Other paper | 1.30 (0.76–2.20); p = 0.340 | - | - | 32.53 (0.59–180.92); p = 0.089 |
| Leaves | 0.88 (0.52–1.46); p = 0.614 | - | - | 0.96 (0.63–1.46); p = 0.842 |
| Nothing | 1.81 (0.35–9.31); p = 0.478 | - | - | 0.86 (0.54–1.35); p = 0.504 |
| **SCHOOL LEVEL FACTORS** | | | | |
| **Drinking water usually available** | | | | |
| Always | - | Reference | - | - |
| Sometimes | - | 1.12 (0.67–1.89); p = 0.658 | - | - |
| Never | - | 2.57 (0.97–6.81); p = 0.058 | - | - |
| **Washed hands with soap and water at Handwashing place the last time they defecated** | | | | |
| Yes | Reference | Reference | - | - |
| No | 1.91 (1.13–3.20); p = **0.015*** | 2.06 (1.21–3.50); **p = 0.008*** | - | - |
| **Drinking water usually available** | | | | |
| Always | Reference | - | - | - |
| Sometimes | 0.93 (0.56–1.56); p = 0.789 | - | - | - |
| **Usually used the drinking water source** | | | | |
| Always | Reference | - | - | Reference |
| Sometimes | 1.25 (0.78–1.99); p = 0.356 | - | - | 1.16 (0.81–1.66); p = 0.422 |
| Never | 0.65 (0.17–2.49); p = 0.525 | - | - | 0.13 (0.02–0.95)**; p = 0.044*** |

*(Continued)*

**Table 5.** (Continued)

| Factors | aOR (95%CI); p-value | | | |
|---|---|---|---|---|
| | Any STH (n = 571) | A.lumbricoides (n = 464) | Hookworm (n = 5) | T. trichiura (n = 212) |
| **School usually provided of anal wiping material** | | | | |
| Always | - | - | - | **Reference** |
| Sometimes | - | - | - | 0.51 (0.12–2.21); p = 0.370 |
| Never | - | - | - | 0.36 (0.12–1.04); p = 0.059 |
| **Number of days they missed school in the past 5 school days** | | | | |
| 0 | Reference | Reference | - | Reference |
| 1–3 | 1.05 (0.61–1.81); p = 0.867 | 0.88 (0.50–1.55); p = 0.654 | - | 1.60 (1.05–2.43); **p = 0.027*** |
| > 3 | 1.89 (0.39–9.18); p = 0.432 | 1.72 (0.30–9.83); p = 0.539 | - | 1.25 (0.45–3.50); p = 0.672 |
| **Type of shoes the child was wearing** | | | | |
| Closed shoes | - | Reference | Reference | - |
| Sandals | - | 1.17 (0.70–1.97); p = 0.542 | 0.78 (0.08–7.81); p = 0.832 | - |
| No Shoes | - | 3.70 (0.97–14.01); p = 0.055 | 7.46 (0.69–80.49); p = 0.098 | - |

*significant factors p<0.05-variables were not included for the multivariable model for infection

## Treatment of school children only and leaving out the rest of the community members

Majority of the study participants felt that school children were being dewormed in school, while leaving out the rest of the community members who could be harbouring the parasites too. Hence, hindering the prevention and control of STH as community members and contaminated environment with faecal matter could be reservoirs of these parasitic diseases. Their opinion was to treat everyone in the endemic communities. Below are selected suggestions from the transcripts:

**P1:** *"My thoughts are as you give our children deworming tablets, why don't you also give us, so we can also be dewormed. Like myself, I need to be dewormed because it is also a problem I am facing"*

**KS-NB-FEMALE FGD**.

**P6:** *"If you deworm the children alone and leave their parents, for these worms to reduce even the parents should be dewormed"*

**KS-EB-MALE FGD**.

## Frequency of annual MDA

Most of the study participants felt that the annual treatment was not adequate to deal with the serious problem of worms. They mentioned that effective control of the parasites can be achieved by frequent deworming. The current recommended WHO treatment schedule is once or twice annual administration which is usually determined by the initial prevalence of

infection with any STH. Their recommendation was bi-annual or quarterly treatment to the school children. Below are extracts of some of the recommendations from the transcripts:

> **P3**: *"In my opinion, once a year will not be enough because that same drugs when you go to the hospital it is given after every three months. So, let us say for example this year they were given in January, then again until January. What about all these other times how will they be?"*
>
> **KS-NS-FEMALE FGD**

> **P5**: *"We need to give children deworming tablets every three months, so we can prevent these worms and we should also boil water for children to drink"*
>
> **KS-NB-FEMALE FGD**.

## Health promotion

Study participants felt that poor knowledge on the importance of owning and using a pit latrine and low personal hygiene was contributing to continuing worm infection among the community members. They suggested the need for health promotion for increased knowledge on good personal hygiene, causes of worm, and prevention and control. Below are some extracts from transcripts:

> **P2**: *"Village elders like the one who was here, the chief, people from the health department, should come to meetings like this, so they can explain to people the consequences of not having toilets."*
>
> **NK-KR-MALE FGD**.

> **P5**: *"When doctors come, you know the Maasai's are illiterate, so you need to call them since they need to be educated. You talk to them anything concerning the worms because they don't know."*
>
> **NK-KR-FEMALE FGD**.

> **P6**: *Even if you give the children the drugs and still there is low hygiene it's still nothing. So, you need to call parents and educate them on hygiene.*
>
> **NK-KR-FEMALE FGD**.

## Discussion

The results of this study have demonstrated that there was moderate prevalence of STH infection at 30.8% and mean intensity was 3875 epg after seven years of consecutive annual MDA. Studies conducted elsewhere in the delta region of Myanmar and Indonesia have shown similar results of STH infection despite continuing annual MDA [30,31]. This could be due to poor sanitation which could be causing environmental contamination with human faecal matter hence enabling re-infection [32].

Results from the qualitative data showed that there were poor WASH conditions both at school and home. In addition, not owning and properly using a pit latrine at home due to various reasons like poverty and lack of awareness to construct and maintain basic toilets

encourages open defecation and environmental contamination with fecal material. This would have been a contributing factor to the slow decline of worms due to food crop contamination with fecal matter hence re-infection with STH. Current WHO guidelines for STH control include MDA programs based on prevalence measurements, aiming at primarily reducing morbidity in school children by lowering the prevalence and intensity of infections to below 1% [33]. This calls for concerted effort towards the control of STH infection among infected SAC and communities by improving the WASH facilities, health promotion and continued MDA.

The most prevalent STH species was *A. lumbricoides* with 24.9%, followed by *T. trichiura* at 11.4% and hookworm at 0.3%. This could be due to most helminths are relatively long-lived in the human host while producing large numbers of environ-mentally resistant eggs voided in the faeces to increase chances of survival and continuity of the lifecycle [34]. More so, *A. lumbricoides* and *T. trichiura* are mainly transmitted through ingestion of contaminated food crops with human faeces, unlike hookworm which infects by penetrating the feet hence reduced infections by wearing shoes or gum boots while in the farm [35]. In addition, environmental conditions like warm weather and soil type could also be contributing to the variation of species-specific infections in the selected counties [36]. Prior studies have shown that lack of safe water and proper sanitation pose a higher risk of STH infections that is distinct according to the route of entry to the human host used by each of the STH species [37]. Qualitative results indicated that annual MDA could be insufficient for control of STH. The study participants suggested biannual MDA for effective control of STH. It is also worth noting that other studies have shown that the three species of STH are effectively treated by albendazole with conflicting theories of alternative treatment strategies due to low efficacy [35,38]. Hence the need to consider bi-annual treatment for effective treatment and control of *A. lumbricoides* and combination of albendazole with mebendazole for *T. trichiura* [12,39].

From the study results, prevalence of STH was highest in Vihiga County at 40.7%, followed by Narok at 34.3% and least in Kisii at 13.8%. Vihiga County also had the highest mean intensity, followed by Narok and Kisii. Previous studies have also shown high prevalence of STH in schools in the Western region of Kenya compared to other parts of the country [9,10]. This could be due to inadequate WASH facilities both in school and at home. It is worth to note that warm weather in this regions could be a conducive habitat for this parasitic infection considering that STH eggs remain viable in warm moist soil conditions. Meanwhile, results from qualitative data showed that most study participants felt that school children were being dewormed in school, while leaving out the rest of the community members who could be harbouring the parasites too. Their opinion was to treat everyone in the endemic communities. Earlier studies have shown that parents could be harbouring this parasites and act as reservoir for re-infecting school children hence hindering the success of MDA programmes [40]. A study conducted by Campbell and colleagues demonstrated the effect that inadequate WASH facilities have on worm burden and emphasized on the need to have an integrated approach in the control and elimination of STH (31,32).

Among the schools, STH prevalence was highest in Ebwiranyi, followed by Nkarano and least in Nyambogo. Kenyan MoE recommends that all learning institutions should have adequate, clean and well-maintained toilets at a ratio of 1 latrine to 25 girls and 1 latrine to 30 boys with a urinal [41]. During the visits to the schools, we observed that some of the schools had a relatively higher population against the available latrine facilities, which were also in poor condition, although this was not measured in structured method in this study. Studies have shown inadequate latrine facilities is associated with STH infection [42]. While in Narok County, a similar situation was witnessed, with majority of the school children reporting poor latrine coverage at home. Qualitative results showed that majority of primary schools had

inadequate latrine for the children and the few which were available were dirty and in poor condition.

Significant associations with increased odds of STH infection were shown in children from Narok and Vihiga. Meaning that the geographical location of these schools could be a factor influencing the high STH infection. This could be due to geographical variations, life style of the community members, soil type/humidity and exposure to contaminated environments. In addition, this are agricultural areas, that are well endowed with fruits, vegetables, and there is a probability of children eating raw food without washing, hence being a possible route for STH (specifically *A. lumbricoides* and *T. trichiura*) infection. Other studies conducted previously have shown that soil type and environmental factors can easily be associated with STH infection [43]. This could be due to other underlying factors like poor sanitation where communities practice open defecation, lack of safe drinking water, crop farming whereby the environment is contaminated with faecal matter and soil type. Acka and colleagues reported a similar practice of poor hygiene of villagers rarely using latrine and defecated where convenient [44]. This practice allows helminths eggs from the feces of infected persons to contaminate the environment including water sources and subsequently infecting the community members. Results from qualitative data showed the need for health promotion among the community members for improved knowledge on the importance of latrine usage and latrine construction, maintenance and usage. In this regard, improving overall sustained access and use of toilets could reduce open defecation and thus reduce STH infection among school age children. In addition, local community members should be sensitized on the importance of owning and using a toilet.

Lack of handwashing with soap and water after defecation was a proxy indicator of broader personal hygiene/handwashing practices, although handwashing after defecation is not considered to be a preventive measure against STH infection due to the need for a maturation period for eggs in the soil before they become infective, lack of handwashing after defecation may indicate overall lack of good hygiene practices, such as before eating/preparing food. This agrees with previous studies conducted elsewhere in Ethiopia which showed that inadequate WASH facilities and poor personal hygiene are closely linked to STH infections [45].

Increased odds of infection with *A. lumbricoides* was significantly associated with Narok and Vihiga counties. This could be due to the farming activities in these counties, the soil type is red which is good for farming activities. Nevertheless, it is worth noting that Kisii County which had the lowest *A. lumbricoides* infection had high latrine coverage compared to the other two counties as per the MoE's requirements. This agrees with a study done elsewhere which showed that factors associated with *A. lumbricoides* infection were environmental variables, particularly alkaline soil and elevation above sea level [42]. In addition, other studies by Vercruysse and colleagues have showed that treatment frequency may lead to drug pressure, less cure rate and possible drug resistance [46].

Lack of connection to electric power supply and not owning household items like sofa set had increased odds of infection. Connection to electric power supply and ownership of household items like sofa set can be an indication of high socio-economic status. Several studies have linked low socio-economic status with increased helminths infection [47,48]. Helminths infections are diseases of poverty, school children from high socio-economic status can afford to get treatment whenever they become ill and to practice good personal hygiene and have a cleaner environment including toilets at home.

Hookworm infections were low and therefore insufficient observations affected the convergence of the model. Nonetheless, children who reported lack of a toilet/latrine at home had increased odds of STH infection. Not owning a toilet/latrine increases like hood of open defecation and environmental contamination, hence increased transmission of STH. Other studies

done in Brazil and Sri Lanka have shown that not owning a toilet/pit latrine might be a contributing factor to the increased helminths infection [36,49]. Failure to take deworming drugs in the previous year was also significantly associated with STH infections. Similar studies done elsewhere have shown that a single-dose oral albendazole was unlikely to be satisfactory for treatment of these parasites [46]. However, other studies have shown that double-dose albendazole or combination of albendazole with ivermectin is more effective for treatment of *T. Trichiura* which Kenya can consider in future MDAs [39,50].

## Study limitations

One of the study limitations was that we used the WHO recommended Kato Katz technique which can miss some egg count in an area with low intensity of infection especially after several rounds of MDA. We recommend the need for more sensitive diagnostic techniques like MC Master for programmatic monitoring for STH in future [51]. While for the qualitative arm of the study, the results cannot be generalizable to other parts of the country as the study was exploring the factors that have contributed to slow decline of STH infection in specific counties. In addition, lack of structured observations of water supply, sanitation infrastructure and associated behaviors was a major limitation to the study due to limited funding which could have improved the study outcomes. Other limitations were failure to triangulate findings on WASH with secondary data on access to water supply and sanitation services, not measuring compliance and logistical issues of the MDA. Finally, failure to examine all transmission pathways such as food crop contamination, testing overall environmental contamination, measuring other conducive aspects such as soil humidity were also limitations of the study.

## Conclusions

This study demonstrated that STH is still a public health problem to school children even after several rounds of continued annual MDA. This calls for continued effort towards annual MDA, community wide treatment, and if possible consider drug combination and bi-annual treatment plans for effective control of morbidities associated with STH. Some of the associated risks of infection were failure to use a pit latrine/toilet at home and school geographical location. Hence, more research is needed to understand the causes of persistent high prevalence of STH infection in the selected counties.

## Acknowledgments

The authors are very grateful to the school administration, school children in the various selected schools and the communities for participation in the study. Also, much appreciation to the County governments of Narok, Kisii and Vihiga especially the Directors of Health and Education for their much support in coordination of the study. Special thanks to all the sub-county public health officers (SPHOs) and laboratory personnel from the three counties for their tireless effort in the data collection process. Finally, thank you to the entire team at Colozzy Data Analytics and Research Solutions specifically to Lilian Owino and Melvine Madoro for coordinating data cleaning and management.

## Author Contributions

**Conceptualization:** Janet Masaku, Collins Okoyo, Sammy M. Njenga.

**Data curation:** Janet Masaku, Collins Okoyo, Sylvie Araka, Rosemary Musuva, Elizabeth Njambi, Doris W. Njomo, Charles Mwandawiro, Sammy M. Njenga.

**Formal analysis:** Janet Masaku, Collins Okoyo.

**Funding acquisition:** Janet Masaku, Collins Okoyo.

**Investigation:** Janet Masaku, Collins Okoyo, Sylvie Araka, Rosemary Musuva, Elizabeth Njambi, Doris W. Njomo, Charles Mwandawiro.

**Methodology:** Janet Masaku, Collins Okoyo, Sylvie Araka, Rosemary Musuva, Elizabeth Njambi, Doris W. Njomo, Charles Mwandawiro.

**Project administration:** Janet Masaku, Collins Okoyo.

**Resources:** Janet Masaku, Collins Okoyo.

**Software:** Janet Masaku, Collins Okoyo.

**Supervision:** Janet Masaku, Collins Okoyo, Sammy M. Njenga.

**Validation:** Janet Masaku, Collins Okoyo, Charles Mwandawiro.

**Visualization:** Janet Masaku, Collins Okoyo, Sammy M. Njenga.

**Writing – original draft:** Janet Masaku, Collins Okoyo, Sylvie Araka, Rosemary Musuva, Elizabeth Njambi, Doris W. Njomo, Charles Mwandawiro, Sammy M. Njenga.

**Writing – review & editing:** Janet Masaku, Collins Okoyo, Doris W. Njomo, Charles Mwandawiro, Sammy M. Njenga.

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
