## [Decision Letter · Decision Letter 0]

3 Jun 2022

Dear Mrs Masaku,

Thank you very much for submitting your manuscript "Understanding factors responsible for the slow decline of soil-transmitted helminths following seven rounds of annual mass drug administration (2012 – 2018) among school age children in endemic counties of Kenya: A mixed method study" for consideration at PLOS Neglected Tropical Diseases. As with all papers reviewed by the journal, your manuscript was reviewed by members of the editorial board and by several independent reviewers. In light of the reviews (below this email), we would like to invite the resubmission of a significantly-revised version that takes into account the reviewers' comments.

We cannot make any decision about publication until we have seen the revised manuscript and your response to the reviewers' comments. Your revised manuscript is also likely to be sent to reviewers for further evaluation.

Sincerely,

Amadou Garba

Associate Editor

Justin Remais

Deputy Editor

Reviewer's Responses to Questions

**Key Review Criteria Required for Acceptance?**

**Methods**

-Are the objectives of the study clearly articulated with a clear testable hypothesis stated?

-Is the study design appropriate to address the stated objectives?

-Is the population clearly described and appropriate for the hypothesis being tested?

-Is the sample size sufficient to ensure adequate power to address the hypothesis being tested?

-Were correct statistical analysis used to support conclusions?

-Are there concerns about ethical or regulatory requirements being met?

Reviewer #1: - The objectives of the study can be discerned from the background section but could be more clearly presented. A testable hypothesis was not provided. 

- The choice of cross-sectional study is appropriate to the subject. However, the design does not seem to have been developed based on a clear conceptual framework of the transmission of STH and the links with WASH. This should ideally be presented in the background/introduction section, including the following elements: 

* Set out the mode of transmission of STH, explaining clearly the need for egg maturation in the soil and the difference with hookworm - this is crucial as it affects the importance of different elements of WASH to each of the STHs. It would also be useful to refer to the resilience of helminth eggs in the environment at this stage (only alluded to much later in the paper) – this is crucial since it can cause significant time lags between improvements in sanitation, and reduced prevalence, which can only be achieved through reducing the overall contamination of the environment. 

* Provide a definition of WASH (noting that this concept includes a vast array of interventions and outcomes, but it is being used in a monolithic way throughout the paper). The authors are advised to revisit the WHO Guidelines on Sanitation and Health for a more detailed examination of the sanitation service chain and its relationship with pathogen transmission, since toilet presence and open defecation are only two potential factors. 

The authors seem to have made certain assumptions about the routes of transmission, which has left out important aspects such as food crop contamination. This has also made it difficult for the authors to make clear links between the behaviours and characteristics included in the study, and the risk of STH transmission. For instance, while handwashing with soap after toilet use may be a useful proxy indicator of handwashing practices, it is less likely to prevent infection than, for instance, handwashing before eating (since the infective eggs on hands are more likely to come from the environment than from contaminated toilet surfaces, as the eggs there are unlikely to have matured). Similarly, while toilet presence in the household is a useful indicator of open defecation practices, toilet type, or whether or not the toilet is considered improved or not, is an important indicator of the risk of contamination of the environment, and this was not included. 

The authors relied on questionnaires and focus group discussions. It is unclear what efforts were made to triangulate or verify the information gathered. Observations of school sanitation facilities are mentioned in the discussion section, but no observations are detailed in the methods section, or any protocol that may have been used to undertake such observations in a structured or consistent way. This reviewer believes that structured observations play an important part in reducing bias and gathering reliable and subjective data on environmental conditions and behavioural practices (not least because problematic behaviours such as lack of handwashing with soap after toilet use, or open defecation, tend to be underreported and therefore suffer from risk of bias), and would have benefited the robustness of this study. Additionally, it is unclear whether efforts were made to utilise local government data on overall rates of access to improved drinking water and sanitation facilities, and levels of open defecation in the study areas. 

- The population is clearly described and appropriate. 

- It is not possible to comment on the sample size since the calculation has not been provided. This should be added. 

- As far as I can tell, the statistical analysis was correct, but this is not my area of expertise. 

- Given that ethical approval was obtained and that an informed consent process was followed, there are no concerns. However, there are some contradictions in the description of the consent process, which I have detailed in my comments.

Reviewer #2: All appropriate

Reviewer #3: The objectives of the study have been clearly articulated by authors; however the study design has major limitations in being able to address these objectives. There are some major concerns related to the link between the objectives and the methods, including:

1. As a cross-sectional study, the authors can only look at the association between STH prevalence or intensity at the time of measurement and the risk factors in the three counties that have been chosen. Mention of a ‘slow decline’ is misleading as the study does not look at any change in prevalence over time. By design, authors themselves have chosen these three counties for having persistent higher prevalence (but no mention is made of measurements over the seven rounds). 

2. Related to the investigation into the three specific counties, without any comparison group(s), we do not know if the risk factors for STH infection are different than in other STH-endemic areas. The risk factors that have been found are primarily known risk factors related to WASH. A comparison with other areas or some consideration for a control area or group would have strengthened the findings.

3. No cluster analysis was undertaken to account for potential clustering in schools or counties/subcounties. 

4. Regarding the age group, it is unclear why all grades from 1 to 7 were included as STH prevalence can be low in early childhood. Younger children have also benefited from few to no rounds of MDA already (unless they participated in Pre-SAC MDA. This is not clear). It would have been better to focus on the older grades with higher STH prevalence. As is, any subgroup analysis by age cannot be conducted due to low sample sizes. 

5. The main indicator for analysis, results and discussion should be based on individual species intensity rather than prevalence (due to the impact of MDA on reduction in intensity and associated morbidity). Also, combining Ascaris, Trichuris and hookworm into one indicator for prevalence (any STH) may be warranted, but having one mean indicator for intensity for all STH species does not seem like a valid approach (as intensity categories have different meanings for the different species). Based on the results, it seems that Ascaris infection is driving the mean numbers. 

6. There is no discussion on any potential attempt to measure compliance, if these specific children were involved in previous rounds of MDA (unlikely the younger ones have had much opportunity), percentage of school children attending school, how many non-enrolled children were targeted in the interventions and study, if MDA was a standalone intervention or accompanied other interventions or health education (which may increase uptake). These details may provide more insight into if and why the intervention may not yet have achieved the desired results. 

7. Authors state that the questionnaire was administered to all SAC (line 195). It is unclear how children could accurately respond to many of the questions. Questionnaires should have been administered to parents to get more accurate responses. 

Some additional comments:

1. Timeline: it is not clear when the study took place and if results are still current. Were results impacted at all by the pandemic (when coverage decreased in some areas)? The only place timeline is mentioned seems to be related to the timing of the MDA (2012-2018) and related to the data used to inform selection of sites (2017). 

2. Sample size: no indication of sample size calculation and methodology for qualitative study.

**Results**

-Does the analysis presented match the analysis plan?

-Are the results clearly and completely presented?

-Are the figures (Tables, Images) of sufficient quality for clarity?

Reviewer #1: The ability to interpret the results suffers from the same issues highlighted in the methods section. There seems to be confusion throughout this section between various identified factors, and the actual risk of STH infection. For example: 

- Toilet cleanliness in schools is mentioned – this is unlikely to be a direct cause of transmission; however, dirty toilets may cause users to prefer open defecation, which in turn increases environmental contamination. 

- The association between lack of handwashing with soap after toilet use and STH infection is mentioned – but as previously noted is more likely to be a proxy indicators of overall handwashing practices than a direct risk factor

- The relationship between poverty indicators (such as not owning a sofa set) and STH infection is not adequately explained or interpreted

- The association between T. trichiura infection with geographical location and source of drinking water is not explained

As noted in the comments on the methods, a clear conceptual framework of STH transmission and the various links with water supply, sanitation and specific hygiene behaviours would have, in my view, made it easier to organise and interpret the results.

Reviewer #2: appropriate

Reviewer #3: The analyses presented do match the analysis plan; however there are major limitations to presentation and interpretation of the results based on the same limitations previously outlined in the methodology. 

For figures:

1. Ideally there should be a figure/diagram to show the flow of participants. Rather than the current Fig. 1, this would be better expressed as a figure showing the actual study/participant flow (in the results section) and not the proposed flow/methodology (as part of the methods. This can still be included in written form).

**Conclusions**

-Are the conclusions supported by the data presented?

-Are the limitations of analysis clearly described?

-Do the authors discuss how these data can be helpful to advance our understanding of the topic under study?

-Is public health relevance addressed?

Reviewer #1: - At the start of the discussion, the risk of reinfection could be expanded upon by referencing the systematic review: Jia TW, Melville S, Utzinger J, King CH, Zhou XN. Soil-transmitted helminth reinfection after drug treatment: a systematic review and meta-analysis. PLoS Negl Trop Dis. 2012;6(5):e1621. doi: 10.1371/journal.pntd.0001621. Epub 2012 May 8. PMID: 22590656; PMCID: PMC3348161.

- The issue of confusing toilet conditions/cleanliness with the risk of infection emerges in the discussion several times – again, it would be important to explain the relationship between this and the risk of infection: for instance, dirty toilets can lead to people preferring to practice open defecation; or poor state of repair of the infrastructure can result in faecal sludge leaking into the environment. 

- Study limitations should be broadened to include several additional aspects: 

o Lack of structured observations

o Lack of triangulation with other information sources/ secondary data 

o Lack of examination of all transmission pathways such as food crop contamination, testing overall environmental contamination, measuring other conducive aspects such as soil humidity, temperature etc.

While there are valid reasons for not including this aspects (mainly cost implications presumably), they should be set out as limitations nonetheless. 

- It is unclear what the suggestion of using CLTS (row 448) is based upon. CLTS is designed to stop open defecation, but on its own does not necessarily lead to the construction of improved facilities that safely separate humans from faeces, and that are more likely to be maintained and sustained over time. It is only the starting point and should not be treated as a silver bullet (see WHO Guidelines on Sanitation and Health, Chapter 5). Approaches to improving overall sustained access and use should be based on the local context. 

- The explanation of the link between lack of handwashing with soap after defecation (rows 448-451) is not biologically plausible, given the need for soil maturation of the eggs for faecal-oral transmission of ascaris and trichuris. As noted, this association is not likely to be indicative of causality, but rather a proxy of overall handwashing practices. 

- While I agree with the final conclusion, I would suggest a deeper understanding is still needed of the causes for persistently high prevalence in the study area, in order to inform the integrated control measures advocated by the authors.

Reviewer #2: appropriate

Reviewer #3: Due to the limitations listed above in the study design, the conclusions are limited and not supported directly by the data as presented. The potential applicability and generalizability of the conclusions are also limited. Authors have stated very few limitations of the study.

**Editorial and Data Presentation Modifications?**

Reviewer #1: Revisions suggested by row number: 

60: Delete 'Helminth' from start of sentence (redundant)

61: Targeted by the World Health Organization. Define EPHP. 

63: the STHs now include strongyloides. Either include here or explain that the paper focuses only on the other three. 

71: Ensure consistency in reference to road map targets (morbidity control mentioned here while eariler, the reference was to EPHP)

73-74: sentence unclear

76: Plese explain the predictive maps being referred to

81: mention prevalence threshold for MDAs

86: please clarify how many of the 66 sub-counties mentioned in the previous paragraph are covered. 

90: replace "infection" with "prevalence"

90-91: mention that 7 rounds of MDA took place during 2012-2018. Please mention the coverage level achieved by those MDAs.

92-93: please swap this sentence with the previous sentence. 

96-97: unclear whether the prevalence levels are for all STH. If there is a breakdown of the species, please provide this in table form. 

102: the strategy period was 2015-2020, not 2015-2022. It has now been succeeded by a renewed strategy 2021-2030 (same timeframe as the NTD road map). Please provide a brief summary of the strategic objectives. 

104: The sentence about the WASH and NTDs sectors is unclear. You could improve this by specifying that the opportunities described can be realised when there is collaboration between those two sectors, including raising awareness, using data for decision making, increasing the evidence base for action, and joint planning.

107: "Primary prevention strategies" are mentioned but not clearly set out.

109-110: setting targets is only one aspect - ultimately, what is needed are interventions to improve access to water supply and sanitation services - and improvement of other preventive practices, noting that STH infection can result from contaminated crops as well. 

119 onwards (study area section): Please be consistent in terms of the information provided for each of the counties. (e.g. number of households and population density). A map would be helpful. 

127: what proportion of the population practices pastoralism? This has a very important bearing on aspects such as open defecation. 

141: replace "where" with "were"

142: What type of M&E report? Coverage? Impact? Prevalence? Replace "where" with "were". 

149: Who participated in the community meetings? 

173: Please include age range 

185: Please include the name of the hospital/s

190: please mention the number of technologists. Please write Ministry of Health in full. 

194: Please provide detail on questionnaire pretesting

205: Please summarise what information was gathered through FGDs, or at least the purpose of the FGDs. 

216-217: Please mention that consent was sometimes obtained from parents/guardians rather than directly from child participants (mentioned earlier). 

218: Sentence beginning with "assent" redundant? 

246: replace "and" with "from the"?

300: "poor WASH conditions both at school and at home" should be broken down: what is the access and coverage of improved household and school toilets? what is the access and coverage of improved drinking water supplies (improved)? And if available: handwashing facilities with water and soap. 

316: the word "ignorance" is perhaps not the right one to use here, as it is value laden. Consider referring to lack of awareness of or capacity to construct and maintain basic toilets. 

346: Perhaps refer to the broader community rather than just the parents are reservoirs. Additionally, the main reservoir of risk of infection is more likely the environment itself given egg resilience. 

354: Frequency of annual MDA - you may wish to refer to the WHO or national treatment guidelines. 

366: Consider referring to health promotion rather than health education. This is a more up to date view of how behaviour change may be achieved. 

390: reconsider the use of "ignorance" as mentioned above. 

408: you may wish to refer to the need for alternative treatment strategies to reduce the risk of resistance. 

416-417: please clarify that eggs remain viable for longer in warm, moist soil conditions. 

418: replace "where" with "were"

442: Please clarify what is meant by "farming activities" - is this a reference to contamination of crops with faecal matter? 

459-461: the last sentence of the paragraph is misplaced. 

487-488: the final sentence is unclear.

Reviewer #2: (No Response)

Reviewer #3: The article would benefit from overall editorial review to improve language, clarity and conciseness.

**Summary and General Comments**

Reviewer #1: This is an important study. It provides valuable impetus to addressing the weaknesses of MDAs as a disease elimination strategy as opposed to disease control imperatives, and highlights the need for increased collaboration between the WASH and NTDs sectors. However, it should be significantly strengthened as suggested in the detailed comments to ensure that it provides a clearer and more robust interpretation of the current situation in Kenya; define more specifically the role of water, sanitation and hygiene interventions; and set out a clear way forward in achieving both national and global NTD targets. 

The most fundamental of my concerns refers to the lack of a conceptual framework that sets out clearly the transmission pathway of STH; as noted, the study has not addressed all pathways, and has not explored in depth the relationship of water, sanitation and hygiene aspects with each pathway. This inevitably leads to a narrow view of the measures needed to achieve elimination, which in turn can undermine efforts to improve cross sectoral collaboration and funding. While the study can of course not be amended retrospectively, I believe that presenting the conceptual framework and using this to interpret the results as well as set out the agenda for further research would greatly improve the paper.

Reviewer #2: Reviewer’s Comments________________________________________

MS Ref.No.: PNTD-D-21-01677

Reviewer: Professor Berhanu Erko

Title: Understanding factors responsible for the slow decline of soil-transmitted helminths following seven rounds of annual mass drug administration (2012 – 2018) among school age children in endemic counties of Kenya: A mixed method study

General Comments: The paper reports the factors that could be influencing the slow decline of STH infections among school children after seven rounds of MDA in endemic counties of Kenya. It has addressed important problem and has also recommended implementation of integrated control efforts of STH by inclusion of WASH programmes, health education for awareness creation on behavior change communication. The paper is suitable for publication in PLOS Neglected Tropical Diseases after the authors have addressed the following specific comments:-

Specific comments

Title:-line 1: It is suggested to replace “soil-transmitted helminths” with “soil-transmitted helminthiasis” in the title as it is more appropriate to refer to the infection rather than the parasite in this regard.

The phrase “school age children” in the title and other parts of the document:- The study participants were selected from enrolled primary schools and were, therefore, “school children”. On the other hand, the authors referred to the study participants as “school age children (SAC)” when they have appropriately defined school age children as “both enrolled and non-enrolled” in lines 86 &87. Hence, it is appropriate to refer to the study participants as “school children” rather than “school age children (SAC)” throughout the paper. 

Page 2, lines 34 & 35 need re-writing for clarity.

Page 7, lines 141 & 142: replace “where” with “were”

Page 7, lines 143: replace “Each study participants were” with “Each study participant was”

Line 163: Better to rewrite “Fig 1: Study profile” as “Figure 3: Study profile” in the caption and as Fig. in the in-text citation (lines 121 & 279).

Lines 291 &292: the statement that reads “This was echoed by the parents/guardians of the SAC as per the themes below;” needs qualification as there are many things below. To which things below the phrase refers? This also holds true for lines 340 & 341.

The statement in lines 340 & 341 is vague and need re-writing for clarity.

Page 19, lines 398 – 400: Differences in mode of transmission cannot be a good explanation for differences in prevalence of the three STHs in this case as long as the children are in the same environment. In case of hookworms, only wearing shoes can make a difference. 

Page 21, line 464: Better not to provide statistical test results in the discussion section.

Page 22, line 482:re-write the statement that reads “…egg count in an area with low prevalence especially after several rounds…” as “…eggs in an area with low intensity of infection especially after several rounds…”

Line 507: replace “Ethics approval and consent to participate” simply with “Ethical approval” or “Ethics statement” because consent to participate is an aspect of ethical approval.

Lines 512 - 514: Delete the sentence that reads “Verbal consent was also obtained from the study participants by explaining the study purpose, confidentiality, and the procedures to be followed”, because the study participants were children for whom proxy consent is sought from parents or guardians.

Lines 514 - 515: Provide age range as “13 to _” instead of above 13 years.

Reviewer #3: Overall, the authors have conducted a study that has the potential for both local and international relevance as the impact of MDA after several rounds is monitored and adjusted accordingly (scaled up or scaled down). Knowing why there is persistent infection in some areas and not others, and incorporating community input and feedback into the MDA process are important goals. However, as is, the authors are unable to elucidate the reasons for continued STH infection in these areas – only what the current risk factors are. The qualitative results are also interesting but may not translate into direct changes to the interventions. Factors such as compliance, accompanying interventions, logistical details related to the MDA, etc. are needed to better understand the potential shortcomings of these interventions.

PLOS authors have the option to publish the peer review history of their article (what does this mean?). If published, this will include your full peer review and any attached files.

Reviewer #1: No

Reviewer #2: Yes: Berhanu Erko

Reviewer #3: No
---

## [Decision Letter · Decision Letter 1]

4 Oct 2022

Dear Mrs Masaku,

Thank you very much for submitting your manuscript "Understanding factors responsible for the slow decline of soil-transmitted helminthiasis following seven rounds of annual mass drug administration (2012 – 2018) among school age children in endemic counties of Kenya: A mixed method study" for consideration at PLOS Neglected Tropical Diseases. As with all papers reviewed by the journal, your manuscript was reviewed by members of the editorial board and by several independent reviewers. In light of the reviews (below this email), we would like to provide you with one addition opportunity to strengthen the manuscript by addressing remaining critiques. We invite the resubmission of a significantly-revised version that takes into account the reviewers' comments, and responds to each remaining reviewer concern with a detailed response.

We cannot make any decision about publication until we have seen the revised manuscript and your response to the reviewers' comments. Your revised manuscript is also likely to be sent to reviewers for further evaluation.

Sincerely,

Justin V. Remais

Section Editor

Reviewer's Responses to Questions

**Key Review Criteria Required for Acceptance?**

**Methods**

-Are the objectives of the study clearly articulated with a clear testable hypothesis stated?

-Is the study design appropriate to address the stated objectives?

-Is the population clearly described and appropriate for the hypothesis being tested?

-Is the sample size sufficient to ensure adequate power to address the hypothesis being tested?

-Were correct statistical analysis used to support conclusions?

-Are there concerns about ethical or regulatory requirements being met?

Reviewer #1: - A conceptual framework was added following the previous review. However, it does not help address the transmission pathway of STH, which is what was missing in order to be able to guide the study and interpret the results. For instance, the framework does not explain that helminth eggs must mature in soil before they can infect a new host, and does not explain how different WASH aspects affect transmission. As it stands, it is confusing and should be removed unless the authors can provide an alternative. 

- While the authors have added some reference to the contents of the WHO Guidelines on Sanitation and Health, the main point raised in earlier comments was that the release of excreta (and therefore helminth eggs) can occur at every stage of the sanitation service chain, not just through open defecation and whether or not a toilet is present. 

- The authors seem not to have address the issue of the association between handwashing after defecation and transmission and this problem is repeated throughout the paper. To explain further - handwashing after defecating is not likely to reduce transmission of STH since the eggs transmitted through this pathway will not have matured in the soil in order to become infective. Handwashing after defecating should therefore be treated as a proxy of handwashing more geenrally, includling before eating - which is more likely to be the relevant transmission pathway. This is alluded to in the discussion (row 508 onwards) but not sufficiently covered or emphasised in the rest of the paper incuding the conclusion. 

- The constraints faced by the researchers, which meant that structured observations could not be undertaken, are understood and cannot be addressed at this stage. However, these should be more thoroughly descrived in the study limitations.

Reviewer #2: (No Response)

Reviewer #3: Improved from previous version. Responded to reviewers' comments.

**Results**

-Does the analysis presented match the analysis plan?

-Are the results clearly and completely presented?

-Are the figures (Tables, Images) of sufficient quality for clarity?

Reviewer #1: - The issue of the association of handwashing after defecation and infection comes up again here and the authors have not addressed earlier comments that this should not be treated as a direct relationship. It is a proxy of broader hygiene/handwashing practices, in the same way that lacking a sofa set is a proxy indicator of poverty (as the authors themselves have noted). As pointed out, the lack of an appropriate conceptual framework makes the analysis harder to organise and interpret.

Reviewer #2: (No Response)

Reviewer #3: Improved from previous version. Responded to reviewers' comments.

**Conclusions**

-Are the conclusions supported by the data presented?

-Are the limitations of analysis clearly described?

-Do the authors discuss how these data can be helpful to advance our understanding of the topic under study?

-Is public health relevance addressed?

Reviewer #1: - The conclusions are weakened by the rest of the analysis as pointed out above. Since the association with handwashing after defecation is referred to without the necessary caveats pointed out above, and no secondary data was included to explain the overall sanitation conditions in the surveyed districts, it is very hard to draw firm conclusions except for "more research is needed". 

- The limitations should be more thoroughly explained since they can inform the design of the subsequent research the authors are calling for in the conclusions. For instance, "structured observations of water supply and sanitation infrastructure and associated behaviours", instead of just "structured observations".

Reviewer #2: (No Response)

Reviewer #3: Improved from previous version. Responded to reviewers' comments.

**Editorial and Data Presentation Modifications?**

Reviewer #1: - Reference to strongyloides has been added, but this is not addressed by the study, which needs to be noted. 

- Prevalence thresholds for MDA should still be included - as far as I understand, MDA is recommended by WHO when the prevalence is over 20 percent. 

- Health education - please apply the revision to "health promotion" in the rest of the paper.

Reviewer #2: (No Response)

Reviewer #3: (No Response)

**Summary and General Comments**

Reviewer #1: This study raises important issues of interest to the scientific community, especially NTDs. It rightly points out that while poor WASH conditions persist, elimination of STH as a public health promblem, or indeed interruption of transmission, will not be possible. 

Nonetheless, the authors should make every effort to not overstate the findings given the limitations to the study caused by (very legitimate) cost and time constraints. They should therefore carefully set these out as pointed above in order to lay out clear recommendations for future research.

Reviewer #2: (No Response)

Reviewer #3: (No Response)

PLOS authors have the option to publish the peer review history of their article (what does this mean?). If published, this will include your full peer review and any attached files.

Reviewer #1: No

Reviewer #2: Yes: Berhanu Erko

Reviewer #3: No
---

## [Decision Letter · Decision Letter 2]

21 Dec 2022

Dear Mrs Masaku,

Thank you very much for submitting your manuscript "Understanding factors responsible for the slow decline of soil-transmitted helminthiasis following seven rounds of annual mass drug administration (2012 – 2018) among school children in endemic counties of Kenya: A mixed method study" for consideration at PLOS Neglected Tropical Diseases. As with all papers reviewed by the journal, your manuscript was reviewed by members of the editorial board and by several independent reviewers. The reviewers appreciated the attention to an important topic. Based on the reviews, we are likely to accept this manuscript for publication, providing that you modify the manuscript according to the review recommendations. Please attend to the remaining reviewer comments, and prepare and submit your revised manuscript within 30 days. If you anticipate any delay, please let us know the expected resubmission date by replying to this email. 

Sincerely,

Justin V. Remais

Section Editor

Justin Remais

Section Editor

Reviewer's Responses to Questions

**Key Review Criteria Required for Acceptance?**

**Methods**

-Are the objectives of the study clearly articulated with a clear testable hypothesis stated?

-Is the study design appropriate to address the stated objectives?

-Is the population clearly described and appropriate for the hypothesis being tested?

-Is the sample size sufficient to ensure adequate power to address the hypothesis being tested?

-Were correct statistical analysis used to support conclusions?

-Are there concerns about ethical or regulatory requirements being met?

Reviewer #1: No further comments - shortcomings to the study design have been noted in the discussion.

Reviewer #2: (No Response)

**Results**

-Does the analysis presented match the analysis plan?

-Are the results clearly and completely presented?

-Are the figures (Tables, Images) of sufficient quality for clarity?

Reviewer #1: No further comments

Reviewer #2: (No Response)

**Conclusions**

-Are the conclusions supported by the data presented?

-Are the limitations of analysis clearly described?

-Do the authors discuss how these data can be helpful to advance our understanding of the topic under study?

-Is public health relevance addressed?

Reviewer #1: No further comments

Reviewer #2: (No Response)

**Editorial and Data Presentation Modifications?**

Reviewer #1: Row 455: consider changing "treatment" to "control", or "control or elimination"

Row 492-493: replace "hence being a proxy for possible STH infection" with "hence being a possible route for STH (specifically A. lumbricoides and T. trichiura) infection"

Row 500: replace "hence the" with "and subsequently infecting"

Row 501: replace "education" with "promotion"

Row 502: add "latrine construction, maintenance and usage"

Row 503-4: refer to "school age children". Also, sanitation improvements in the community may not prevent infection in the school. 

Row 507-8: Delete the following: "and therefore a possible pathway towards STH infection". Replace with "Although handwashing after defecation is not considered to be a preventive measure against STH infection due to the need for a maturation period for eggs in the soil before they become infective, lack of handwashing after defecation may indicate overall lack of good hygiene practices, such as before eating/preparing food."

Row 508-9: Delete the following: "This could have led to contamination of food, surfaces through touching, and the

environment hence ingesting the eggs and continuation of the helminth lifecycle"

Row 526: add the following at the end of the sentence: "and have a cleaner environment including toilets at home."

Row 548: replace "triangulate secondary data" with "triangulate findings on WASH with secondary data on access to water supply and sanitation services."

Reviewer #2: (No Response)

**Summary and General Comments**

Reviewer #1: No further comments.

Reviewer #2: (No Response)

PLOS authors have the option to publish the peer review history of their article (what does this mean?). If published, this will include your full peer review and any attached files.

Reviewer #1: No

Reviewer #2: Yes: Berhanu Erko

Figure Files:

Data Requirements:

Reproducibility:

References

---

## [Editor Report · Decision Letter 3]

13 Apr 2023

Dear Mrs Masaku,

We are pleased to inform you that your manuscript 'Understanding factors responsible for the slow decline of soil-transmitted helminthiasis following seven rounds of annual mass drug administration (2012 – 2018) among school children in endemic counties of Kenya: A mixed method study' has been provisionally accepted for publication in PLOS Neglected Tropical Diseases.

Please see the recommendations of reviewer #1, which you may choose to adopt as you finalize your manuscript for publication.

Best regards,

Justin V. Remais

Section Editor

---

## [Editor Report · Acceptance letter]

26 Apr 2023

Dear Mrs Masaku,

We are delighted to inform you that your manuscript, "Understanding factors responsible for the slow decline of soil-transmitted helminthiasis following seven rounds of annual mass drug administration (2012 – 2018) among school children in endemic counties of Kenya: A mixed method study," has been formally accepted for publication in PLOS Neglected Tropical Diseases.

Best regards,

Shaden Kamhawi

co-Editor-in-Chief

Paul Brindley

co-Editor-in-Chief
